# Vacuum and Spacetime Signature in the Theory of Superalgebraic Spinors

**Vadim Monakhov** 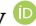

Institute of Physics, Saint Petersburg University, Ulyanovskaya 1, Stariy Petergof,
Saint Petersburg 198504, Russia; v.v.monahov@spbu.ru

**Abstract:** A new formalism involving spinors in theories of spacetime and vacuum is presented. It is based on a superalgebraic formulation of the theory of algebraic spinors. New algebraic structures playing role of Dirac matrices are constructed on the basis of Grassmann variables, which we call gamma operators. Various field theory constructions are defined with use of these structures. We derive formulas for the vacuum state vector. Five operator analogs of five Dirac gamma matrices exist in the superalgebraic approach as well as two additional operator analogs of gamma matrices, which are absent in the theory of Dirac spinors. We prove that there is a relationship between gamma operators and the most important physical operators of the second quantization method: number of particles, energy–momentum and electric charge operators. In addition to them, a series of similar operators are constructed from the creation and annihilation operators, which are Lorentz-invariant analogs of Dirac matrices. However, their physical meaning is not yet clear. We prove that the condition for the existence of spinor vacuum imposes restrictions on possible variants of the signature of the four-dimensional spacetime. It can only be $(1, -1, -1, -1)$, and there are two additional axes corresponding to the inner space of the spinor, with a signature $(-1, -1)$. Developed mathematical formalism allows one to obtain the second quantization operators in a natural way. Gauge transformations arise due to existence of internal degrees of freedom of superalgebraic spinors. These degrees of freedom lead to existence of nontrivial affine connections. Proposed approach opens perspectives for constructing a theory in which the properties of spacetime have the same algebraic nature as the momentum, electromagnetic field and other quantum fields.

**Keywords:** spacetime signature; space-time signature; Clifford algebra; gamma matrices; Clifford vacuum; spinor vacuum; Clifford bundle; spinor bundle

## 1. Introduction

The question of the origin of the dimension and signature of spacetime has a long history. There are different approaches to substantiate the observed dimension and the spacetime signature. One of the main directions is the theory of supergravity. It was shown in [1] that the maximum dimension of spacetime, at which supergravity can be built, is equal to 11. At the same time, multiplets of matter fields for supersymmetric Yang–Mills theories exist only when the dimension of spacetime is less than or equal to 10 [2]. Subsequently, most attention was paid to the theory of superstrings and supermembranes. Various versions of these theories were combined into an 11-dimensional M-theory [3,4]. In [5], the most general properties of the theories of supersymmetry and supergravity in spaces of various dimensions and signatures were analyzed. Proceeding from the possibility of the existence of Majorana and pseudo-Majorana spinors in such spaces, it was shown that the supersymmetry and the supergravity of M-theory can exist in 11-dimensional and 10-dimensional spaces with arbitrary signatures, although, depending on the signature, the theory type differs.

Later, other possibilities were shown for constructing variants of M-theories in spaces of different signatures [6].

Other approaches are Kaluza–Klein theories. For example, in [7], it was shown that, in some cases, it is possible not only to postulate, but also to determine from the dynamics the dimension of the spacetime as well as its signature. In [8–10], an attempt was made to find a signature based on the average value of the quantum fluctuating metric of spacetime.

A number of other attempts were made to explain the dimension and the signature of spacetime. For example, the anthropic principle and causality were used in [11], the existence of equations of motion for fermions and bosons was used in [12] and the possibility of existence in spacetime classical electromagnetism was used in [13].

In all the above approaches, the fermion vacuum operator in the second quantization formalism is not constructed and the restrictions imposed by such a construction are not considered. Therefore, the possibility of the existence of the vacuum and fermions is not discussed.

## 2. Theory of Algebraic Spinors

The most general approach to the theory of spinors is based on the theory of Clifford algebras [14–20]. The corresponding theory was called the theory of algebraic spinors.

The central role in the theory of algebraic spinors is played by the Hermitian primitive idempotent $I$ [18–21]

$$
\begin{aligned}
I &\in \mathbb{C} \otimes Cl(p,q), \\
I^2 &= I, \\
I^+ &= I,
\end{aligned}
\tag{1}
$$

where $\mathbb{C}$ is the field of complex numbers and $Cl(p,q)$ is complex or real Clifford algebra with signature $(p,q)$, being $p$ the number of basis vectors with positive signature and $q$ the negative one.

The subset $M(I) = \{U \in \mathbb{C} \otimes Cl(p,q); U = UI\}$ is called the left ideal generated by $I$. It is a complex vector space which is called spinor space. Elements of this ideal are spinors. They are called algebraic spinors.

All complex Clifford algebra $Cl^{\mathbb{C}}(p,q)$ with $p + q = n$ are isomorphic [20]. We consider only even-dimensional complex Clifford algebras $Cl^{\mathbb{C}}(m,m)$ with the number of basis vectors $n = 2m$. Algebra $Cl^{\mathbb{C}}(m,m)$ has a complete set of $2^m$ primitive mutually annihilating idempotents $I$. Despite the fact that the theory of algebraic spinors has a greater generality compared to the ordinary matrix theory of the Dirac 4-spinors, it has problems with physical interpretation. The matrix representation of algebraic 4-spinors is a $4 \times 4$ matrix, and each column in it is an independent Dirac 4-spinor [16–20]. The space of algebraic spinors represented by such a matrix can contain not only Dirac, Majorana and Weyl spinors, but also the so-called flag-dipole and flagpole spinors [16,22].

The theory of algebraic spinors allows describing fields with different spins [23], which opens the perspective for constructing supersymmetric field theories on its basis. However, for this, odd Grassmann variables must be introduced, and the algebra of spinors and operators acting on them must be transformed into a superalgebra.

## 3. Theory of Superalgebraic Spinors and Vacuum State

There are various approaches that allow adding Grassmann variables to the theory of Clifford algebras. We use the approach in which Grassmann variables are defined in the framework of the classic theory of algebraic spinors where they are constructed of Clifford basis vectors [21,24,25].

Let us consider the even-dimensional $n = 2m$ complex Clifford algebra with basis vectors $E_j$, where $j = 1, 2, \ldots, n$ and $(E_j)^2 = 1$, $E_j^+ = E_j$ [21]. We call this algebra a large Clifford algebra.

Let us introduce

$$\theta^\alpha = \frac{E_{2\alpha-1} - iE_{2\alpha}}{2},$$
$$\overline{\theta^\alpha} = \frac{E_{2\alpha-1} + iE_{2\alpha}}{2},$$

(2)

where $\alpha = 1, 2, \ldots, m$.

From the definition in Equation (2), it follows that

$$(\theta^\alpha)^2 = (\overline{\theta^\alpha})^2 = 0,$$
$$\{\theta^\alpha, \overline{\theta^\beta}\} = \delta^\alpha_\beta.$$

(3)

Equation (3) defines the canonical anticommutation relations—the CAR-algebra of variables $\theta^\alpha$ and $\overline{\theta^\beta}$. Clifford vectors $E_j$ are odd elements of the Clifford algebra. Therefore, variables $\theta^\alpha$ and $\overline{\theta^\beta}$ are also odd elements of the Clifford algebra.

Using these variables, one can construct an Hermitian primitive idempotent [21,24]

$$I_1 = \overline{\theta^1}\theta^1\overline{\theta^2}\theta^2 \ldots \overline{\theta^m}\theta^m.$$

(4)

It plays the role of a spinor vacuum [21], and variables $\theta^\alpha$ and $\overline{\theta^\alpha}$ are the creation and annihilation operators [21].

Other idempotents $I_i$ differ from $I_1$ only in the order of factors $\theta^\alpha$ and $\overline{\theta^\alpha}$. For example,

$$I_2 = \theta^1\overline{\theta^1}\,\overline{\theta^2}\theta^2 \ldots \overline{\theta^m}\theta^m,$$
$$I_3 = \overline{\theta^1}\theta^1\theta^2\overline{\theta^2} \ldots \overline{\theta^m}\theta^m,$$
$$\ldots$$
$$I_{2^m} = \theta^1\overline{\theta^1}\,\theta^2\overline{\theta^2} \ldots \theta^m\overline{\theta^m}.$$

(5)

As already mentioned, there are $2^m$ such idempotents. Each idempotent $I_i$ generates its own spinor space $M(I_i)$, which is the minimal left ideal of the algebra. In turn, each ideal $M(I_i)$ has a basis of $2^m$ elements, and Clifford algebra is the direct sum of these ideals.

For variables $\overline{\theta^\alpha}$, we can use the notation [25]

$$\overline{\theta^\alpha} = \frac{\partial}{\partial\theta^\alpha}.$$

(6)

Taking into account Equation (6), Equation (3) can be rewritten as

$$(\theta^\alpha)^2 = (\frac{\partial}{\partial\theta^\alpha})^2 = 0,$$
$$\{\theta^\alpha, \frac{\partial}{\partial\theta^\beta}\} = \delta^\alpha_\beta.$$

(7)

Equation (7) represents the usual anticommutation relations for Grassmann variables and derivatives with respect to them.

From the previous arguments, it is clear that $\overline{\theta^\alpha}$ can be considered as Grassmann variable with the same success, and $\theta^\alpha$ can be considered as a derivative with respect to $\overline{\theta^\alpha}$:

$$\theta^\alpha = \frac{\partial}{\partial\overline{\theta^\alpha}}.$$

(8)

However, if we consider a particular primitive idempotent $I$, we can always define the Grassmann variables in such a way they serve as creation operators, and, accordingly, the derivatives with respect

to them serve as annihilation operators. Thus, the idempotent would have the form of Equation (4). We denote this idempotent as $\Psi_V$, and the relations for $\Psi_V$ are satisfied

$$
\begin{aligned}
\theta^\alpha \, \Psi_V &\neq 0 \,, \\
\frac{\partial}{\partial \theta^\alpha} \Psi_V &= 0 \,.
\end{aligned}
\tag{9}
$$

The formulation of the theory of algebraic spinors, in which all elements are expressed in terms of Grassmann variables and derivatives with respect to them, is called the theory of superalgebraic spinors.

Apparently, the first studies in the theory of superalgebraic spinors were made by N. Borštnik [26–28]. However, N. Borštnik's interpretation of Grassmann variables and derivatives with respect to them was very different from that proposed in this article and had nothing to do with the theory of algebraic spinors. She assumed that the Grassmann variables are supercoordinates of a superspace, not creation operators of the spinor (fermion). Accordingly, she believed that the derivatives with respect to the Grassmann variables, multiplied by the imaginary unit, are components of the supermomentum, and not operators of the annihilation of the spinor. In these works, formulas for the Lorentz transformations of superalgebraic spinors and a number of other useful results are obtained.

## 4. Superalgebraic Analog of Matrices

The author develops an approach to the theory of superalgebraic spinors, in which Clifford analogs of Dirac gamma matrices are composite.

In [29,30], it was shown that, using Grassmann variables and derivatives with respect to them, one can construct an analog of matrix algebra, including analogs of matrix columns of four spinors and their adjoint rows of conjugate spinors. However, at the same time, the spinors and their conjugates exist in the same space—i.e., in the same algebra.

Initially, this approach was based on the idea of using Grassmann variables and derivatives with respect to them in the spirit of the theory of supersymmetry and was not based on the theory of algebraic spinors. Let us prove the correctness of this approach (with some corrections) within the framework of the theory of algebraic spinors.

Consider the spinor space $M(\Psi_V)$, which is a left ideal generated by the idempotent $\Psi_V$ given by Equation (4). This space is obtained by multiplying $\Psi_V$ on the left by all elements of the Clifford algebra. Therefore, in the Clifford algebra under consideration, any operators transform the elements of a given ideal into its other elements.

Element $\Phi$ of the ideal is called state vector. It can be written as

$$
\Phi = \left( \phi_0 + \phi_{\alpha_{i_1}} \theta^{\alpha_{i_1}} + \phi_{\alpha_{i_1} i_2} \theta^{\alpha_{i_1}} \theta^{\alpha_{i_2}} + \dots \phi_{\alpha_{i_1 \dots i_m}} \theta^{\alpha_{i_1}} \dots \theta^{\alpha_{i_m}} \right) \Psi_V = \phi \Psi_V \,.
\tag{10}
$$

In the transformation, which consists in multiplying the state vector $\Phi$ by the operator $T$

$$
\Phi' = T\Phi \,,
\tag{11}
$$

an arbitrary operator $A$ is transformed as

$$
A' = TAT^{-1} \,.
\tag{12}
$$

Consider an arbitrary element $\Psi$ of a linear vector space $Cl_1^{\mathbb{C}}(m, m)$ with a basis $E_j$, or, equivalently, with a basis as Equation (2). It can be written as

$$
\Psi = \psi^\alpha \frac{\partial}{\partial \theta^\alpha} + \chi_\alpha \theta^\alpha \,,
\tag{13}
$$

where $\alpha = 1, 2, \dots, m$.

Element $\Psi\Phi$ belongs to the same ideal as $\Phi$, and operator $\Psi$ corresponds to the spinor field operator in second quantized field theory.

Suppose there is some arbitrary operator of infinitesimal transformations

$$T = 1 + \varepsilon M \tag{14}$$

where $M$ is any element of the Clifford algebra, and $\varepsilon$ is an infinitely small parameter of the transformation.

In accordance with Equations (11), (12) and (14), it provides infinitesimal transformations (Equation (15)) of $\Phi$ and $A$

$$\begin{aligned}
\Phi' &= \Phi + \varepsilon M \Phi, \\
A' &= A + \varepsilon [M, A].
\end{aligned} \tag{15}$$

and generates for elements $A$ a Lie group corresponding to the Clifford group.

Denote

$$\hat{M} = [M, *], \tag{16}$$

where operator $[M, *]$ means commutator when it acts on $A$ and means multiplying by $M$ when it acts on state vector $\Phi$.

In this case,

$$\begin{aligned}
\hat{M}\Psi &= [M, \Psi], \\
\hat{M}\Phi &= [M, \Phi] = M\Phi, \\
\hat{M}\Psi\Phi &= [M, \Psi]\Phi + \Psi M\Phi.
\end{aligned} \tag{17}$$

Element $M$ of the Lie group is the sum of the Clifford scalar and the Clifford bivector [20]. In the language of the superalgebraic formalism, this means that

$$\hat{M} = \left[ a + b^{\alpha\beta} \frac{\partial}{\partial \theta^\alpha} \frac{\partial}{\partial \theta^\beta} + c^\alpha_\beta \frac{\partial}{\partial \theta^\alpha} \theta^\beta + d_{\alpha\beta} \theta^\alpha \theta^\beta, * \right], \tag{18}$$

where $a, b^{\alpha\beta}, c^\alpha_\beta, d_{\alpha\beta}$ are numerical constants. Moreover, we can assume that $a = 0$ as long as we consider only commutators.

It is easy to show, in complete analogy with the work in [29,30], that operator $\hat{M}$ is a superalgebraic analog of the matrix that transforms column $\Psi$.

Let the value $m$, which specifies the number of Grassmann variables in the large Clifford algebra, be $m = 2\nu$, where $\nu$ is some integer. In this case, the dimension of space of operators $\hat{M}$ which we consider as analogs of matrices is equal to $2^\nu \times 2^\nu$, and it is possible to set $2\nu$ operators (analogs of gamma matrices) $\hat{\gamma}^\mu_p$, which are generators of the corresponding Clifford algebra. For the four-dimensional case, they are given by Equation (A1).

The signature of these operators can be set arbitrary due to the possibility of multiplying any of these "matrices" by $i$. The first four operators $\hat{\gamma}^\mu_p$ correspond to the Dirac gamma matrices $\gamma^\mu$. Operators $\hat{\gamma}^6_p$ and $\hat{\gamma}^7_p$ have no analogs in the Dirac theory. The reason for using the index $p$ is explained below.

We call this algebra the small Clifford algebra. It should be noted that this algebra is Clifford algebra only under field operators $\Psi$ (Equation (13)). When acting on state vectors $\Psi\Phi$, it is necessary to take into account term $\Psi M\Phi$ in Equation (17).

Generators of pseudo-orthogonal rotations are given by the usual relation:

$$\hat{\gamma}^{\mu\nu}_p = \frac{\hat{\gamma}^\mu_p \hat{\gamma}^\nu_p - \hat{\gamma}^\nu_p \hat{\gamma}^\mu_p}{2}. \tag{19}$$

For the four-dimensional case, these operators generate Lorentz transformations. We mean active Lorentz transformations [31] when the basis Clifford vectors and all other vectors are rotated, and at the same time all vectors retain the old coordinates in the new basis.

Consider boost (active Lorentz transformation) of the time-like momentum vector $P = \gamma^0 m$ in the Clifford algebra with generators $\gamma^\mu$, where $m$ is real positive constant. Let the operator of transformation

$$T = exp(\gamma^{0k}\omega_{0k}/2)\,, \tag{20}$$

where real constants $\omega_{0k} = -\omega_{k0}$ are parameters of the transformation and $k = 1, 2, 3$.

Clifford vector $P$ after transformation appears as

$$P' = TPT^{-1} = e^{\gamma^{0k}\omega_{0k}/2}\gamma^0 m\, e^{-\gamma^{0k}\omega_{0k}/2} = e^{\gamma^0\gamma^k\omega_{0k}}\gamma^0 m\,. \tag{21}$$

Let

$$\varphi = \sqrt{\sum_k (\omega_{0k})^2} \tag{22}$$

and

$$\gamma = -\gamma^k\omega_{0k}/\varphi\,, \tag{23}$$

wherein $(\gamma)^2 = -1$.

Then, $\gamma^k\omega_{0k} = -\gamma\varphi$, and we have from Equation (21) that

$$P' = e^{-\gamma^0\gamma\varphi}\gamma^0 m = \gamma^0 m\,cosh(\varphi) - \gamma^0\gamma\gamma^0 m\,sinh(\varphi) = \gamma^0 m\,cosh(\varphi) + \gamma\,m\,sinh(\varphi)\,. \tag{24}$$

Equation (24) can be rewritten as

$$P' = \gamma^0 p_0 + \gamma\,p = \gamma^0 p_0 + \gamma^k p_k\,, \tag{25}$$

where $p_0 = m\,cosh(\varphi)$ and $p_k = -m\,\frac{\omega_{0k}}{\varphi}sinh(\varphi)$.

Taking into account Equations (22) and (25), we find that parameters $\omega_{0k}$ uniquely set spatial momentum $p'$ after the rotation, and $p'$ has components $p_k$.

Moreover, if there is a dependence $A(p)$ of the element $A$ on the momentum $p$, then, after rotation, we obtain the dependence $A'(p')$ of the transformed element $A'$ on $p'$. Therefore, $A(0)$ is transformed to $A'(p')$. For example, if we have density $\theta^\alpha(p)$ (see next section), and initially $p = 0$, then after the rotation we get density $\theta'^\alpha(p')$.

The operator of the Lorentz transformation of a general form, including not only boosts, but also spatial rotations, has the form

$$T = exp(\gamma^{\mu\nu}\omega_{\mu\nu}/4)\,, \tag{26}$$

where real constants $\omega_{\mu\nu} = -\omega_{\nu\mu}$ are parameters of the transformation and in the four-dimensional case $\mu, \nu = 0, 1, 2, 3$. Otherwise, the ranges of $\mu$ and $\nu$ values correspond to all existing Clifford algebra generators.

These arguments are valid for any Clifford algebra, including the small Clifford algebra with generators $\hat{\gamma}_p^{\mu\nu}$.

In this case, the state vector in Equation (10) is transformed as

$$\Phi' = (\phi_0 + \phi_{\alpha_{i_1}}\theta'^{\alpha_{i_1}} + \phi_{\alpha_{i_1 i_2}}\theta'^{\alpha_{i_1}}\theta'^{\alpha_{i_2}} + \ldots \phi_{\alpha_{i_1\ldots i_m}}\theta'^{\alpha_{i_1}}\ldots\theta'^{\alpha_{i_m}})\Psi_V'\,, \tag{27}$$

where $\theta'^\alpha = exp(\hat{\gamma}_p^{\mu\nu}\omega_{\mu\nu}/4)\theta^\alpha$ and $\Psi_V'$ is transformed vacuum state.

All rotations, with the exception of boosts, leave the vacuum invariant. This is because

$$\hat{\gamma}_p^{ab}\Psi_V = 0\,, \tag{28}$$

where $a, b = 1, 2, 3, 4, 6, 7$.

　　Equation (27) corresponds to the relations of quantum field theory for the method of second quantization, if operators $\theta^\alpha$ are considered as the operators of the creation of states with momentum $p = 0$ and $\theta'^\alpha$ as the operators of the creation of states with momentum $p$. In this case, the vacuum state vector must be invariant even with boosts. This requires an approach in which operators $\theta^\alpha$ depend on the momentum.

　　There is another serious reason that requires going beyond the theory of algebraic spinors and small Clifford algebra allows us to do this. In the general case, it is impossible to decompose the elements of the Clifford algebra on manifold into elementary spinors, since an arbitrary manifold does not admit covariantly constant idempotent fields [18]. The proposed approach ensures that spinors of a small Clifford algebra have additional internal degrees of freedom due to the fact that a large Clifford algebra contains more elements than a small one.

## 5. Superalgebraic Analog of Dirac Gamma Matrices and Operators of Pseudo-Orthogonal Rotations

　　In [32,33], the author proposed such approach. Grassmann densities $\theta^a(p)$, $a = 1, 2, 3, 4$, and derivatives $\frac{\partial}{\partial \theta^a(p)}$ with respect to them were introduced, with CAR-algebra

$$\{\frac{\partial}{\partial \theta^i(p)}, \theta^k(p')\} = \delta(p - p')\delta_i^k. \tag{29}$$

　　Operators $\hat{\gamma}^\mu$ (Equation (30)) are constructed of these densities. They are superalgebraic analogs of Dirac gamma matrices $\gamma^\mu$. We call them gamma operators.

$$
\begin{aligned}
\hat{\gamma}^0 &= \int d^3 p \left[\frac{\partial}{\partial \theta^1(p)}\theta^1(p) + \frac{\partial}{\partial \theta^2(p)}\theta^2(p) + \frac{\partial}{\partial \theta^3(p)}\theta^3(p) + \frac{\partial}{\partial \theta^4(p)}\theta^4(p), *\right], \\
\hat{\gamma}^1 &= \int d^3 p \left[\frac{\partial}{\partial \theta^1(p)}\frac{\partial}{\partial \theta^4(p)} - \theta^4(p)\theta^1(p) + \frac{\partial}{\partial \theta^2(p)}\frac{\partial}{\partial \theta^3(p)} - \theta^3(p)\theta^2(p), *\right], \\
\hat{\gamma}^2 &= i \int d^3 p \left[-\frac{\partial}{\partial \theta^1(p)}\frac{\partial}{\partial \theta^4(p)} - \theta^4(p)\theta^1(p) + \frac{\partial}{\partial \theta^2(p)}\frac{\partial}{\partial \theta^3(p)} + \theta^3(p)\theta^2(p), *\right], \\
\hat{\gamma}^3 &= \int d^3 p \left[\frac{\partial}{\partial \theta^1(p)}\frac{\partial}{\partial \theta^3(p)} - \theta^3(p)\theta^1(p) - \frac{\partial}{\partial \theta^2(p)}\frac{\partial}{\partial \theta^4(p)} + \theta^4(p)\theta^2(p), *\right], \\
\hat{\gamma}^4 &= i\hat{\gamma}^5 = i \int d^3 p \left[\frac{\partial}{\partial \theta^1(p)}\frac{\partial}{\partial \theta^3(p)} + \theta^3(p)\theta^1(p) + \frac{\partial}{\partial \theta^2(p)}\frac{\partial}{\partial \theta^4(p)} + \theta^4(p)\theta^2(p), *\right], \\
\hat{\gamma}^6 &= i \int d^3 p \left[\frac{\partial}{\partial \theta^1(p)}\frac{\partial}{\partial \theta^2(p)} + \theta^2(p)\theta^1(p) - \frac{\partial}{\partial \theta^3(p)}\frac{\partial}{\partial \theta^4(p)} - \theta^4(p)\theta^3(p), *\right], \\
\hat{\gamma}^7 &= \int d^3 p \left[\frac{\partial}{\partial \theta^1(p)}\frac{\partial}{\partial \theta^2(p)} - \theta^2(p)\theta^1(p) + \frac{\partial}{\partial \theta^3(p)}\frac{\partial}{\partial \theta^4(p)} - \theta^4(p)\theta^3(p), *\right].
\end{aligned}
\tag{30}
$$

　　In contrast to the previous approach, the operator $[M, *]$ is always considered as a commutator. This does not affect the relations of the small Clifford algebra, but it allows ensuring the invariance of the vacuum with boosts.

　　The theory is automatically secondarily quantized and does not require normalization of operators.

　　In the proposed theory, in addition to analogs of the Dirac matrices, there are two additional gamma operators $\hat{\gamma}^6$ and $\hat{\gamma}^7$, the rotation operator in whose plane (gauge transformation) is analogous to the charge operator of the second quantization method [33].

　　In [33], the superalgebraic analog of the Dirac conjugation was proposed. It was shown that the general form of the conjugation that provides Lorentz covariance is given by the Equation

$$
\begin{aligned}
\overline{\Psi} &= (M_{p,q}\Psi)^+, \\
M_{p,q} &= \hat{\gamma}^1_+ \ldots \hat{\gamma}^p_+ c_+ + \hat{\gamma}^1_- \ldots \hat{\gamma}^q_- c_-,
\end{aligned}
\tag{31}
$$

where $c_+$ and $c_-$ are numerical constants and $(p, q)$ is signature of spacetime, $\hat{\gamma}^i_+$—gamma operator with positive signature—and $\hat{\gamma}^i_-$—gamma operator with negative signature. The signature of the spacetime sets the formula for the Dirac conjugation, and vice versa.

In [33], it was shown that transformations of densities $\theta^a(p)$ and $\frac{\partial}{\partial\theta^a(p)}$, while maintaining their CAR-algebra of creation and annihilation operators, provide transformations of field operators of the form

$$\Psi' = (1 + i\hat{\gamma}^\mu d\omega_\mu + \frac{1}{4}\hat{\gamma}^{\mu\nu} d\omega_{\mu\nu})\Psi,\tag{32}$$

where $\hat{\gamma}^{\mu\nu} = \frac{1}{2}(\hat{\gamma}^\mu\hat{\gamma}^\nu - \hat{\gamma}^\nu\hat{\gamma}^\mu)$; $\mu, \nu = 0, 1, 2, 3, 4, 6, 7$, and $d\omega_{\mu\nu} = -d\omega_{\nu\mu}$ are the real infinitesimal transformation parameters. The multiplier $1/4$ is added in Equation (32) compared to the work in [33] to correspond to the usual transformation formulas for spinors in the case of Lorentz transformations.

Operators $\hat{\gamma}^{\mu\nu}$ are the generators of the pseudo-orthogonal rotations of the form $exp(\hat{\gamma}^{\mu\nu}\omega_{\mu\nu}/4)$, where $\mu, \nu = 0, 1, 2, 3, 4, 6, 7$. We call them gamma operators of rotations. They are the generators of Lorentz rotations when $\mu, \nu = 0, 1, 2, 3$.

Operators of annihilation of spinors $b_\alpha(p)$, $\alpha = 1, 2$, and of antispinors $b_\tau(p)$, $\tau = 3, 4$, are obtained by Lorentz rotations of $\frac{\partial}{\partial\theta^\alpha(0)}$ and $\frac{\partial}{\partial\theta^\tau(0)}$, and the Dirac conjugated to the operators of creation $\bar{b}_\alpha(p)$ and $\bar{b}_\tau(p)$ by Lorentz rotations of $\theta^\alpha(0)$ and $\theta^\tau(0)$ [32,33]. The momentum specified as a parameter is replaced with a rotation from 0 to $p$:

$$b_i(p) = (e^{\hat{\gamma}^{0k}\omega_{0k}/2}\frac{\partial}{\partial\theta^i(0)})|_{0\to p},$$
$$\bar{b}_i(p) = (e^{\hat{\gamma}^{0k}\omega_{0k}/2}\theta^i(0))|_{0\to p},\tag{33}$$
$$i = 1, 2, 3, 4.$$

Anticommutation relations for $b_i(p)$ and $\bar{b}_k(p')$

$$\{b_i(p), \bar{b}_k(p')\} = \delta(p - p')\delta^k_i.\tag{34}$$

In Equation (33), the particle momentum $p$ depends on Lorentz rotation parameters $\omega_{0k}$. For example, for rotation in the plane $\hat{\gamma}^0$, $\hat{\gamma}^1$, the transformation (33) for $b_1(p)$ and $\bar{b}_1(p)$ looks like

$$b_1(p) = cosh\frac{\omega_{01}}{2}\frac{\partial}{\partial\theta^1(p)} + sinh\frac{\omega_{01}}{2}\hat{\gamma}^{01}\frac{\partial}{\partial\theta^1(p)},$$
$$\bar{b}_1(p) = cosh\frac{\omega_{01}}{2}\theta^1(p) + sinh\frac{\omega_{01}}{2}\hat{\gamma}^{01}\theta^1(p).\tag{35}$$

As a result, we get

$$b_1(p) = cosh\frac{\omega_{01}}{2}\frac{\partial}{\partial\theta^1(p)} + sinh\frac{\omega_{01}}{2}\theta^4(p),$$
$$\bar{b}_1(p) = cosh\frac{\omega_{01}}{2}\theta^1(p) - sinh\frac{\omega_{01}}{2}\frac{\partial}{\partial\theta^4(p)}.\tag{36}$$

The expression for operator $\hat{\gamma}^{01}$ as an example of the operator of rotations is given by

$$\hat{\gamma}^{01} = \int d^3p\,[\frac{\partial}{\partial\theta^1(p)}\frac{\partial}{\partial\theta^4(p)} + \theta^4(p)\theta^1(p) + \frac{\partial}{\partial\theta^2(p)}\frac{\partial}{\partial\theta^3(p)} + \theta^3(p)\theta^2(p), *].\tag{37}$$

Similar expressions for all gamma operators of rotations $\hat{\gamma}^{ab}$ are given by Equation (A3).

Denote the integrands in Equation (A2) as $\hat{\gamma}^a(p)$ and in Equation (A3) as $\hat{\gamma}^{ab}(p)$. Thus, we can rewrite Equation (A2) as

$$\hat{\gamma}^a = \int d^3p\,\hat{\gamma}^a(p),\tag{38}$$

and (A3) as

$$\hat{\gamma}^{ab} = \int d^3p \, \hat{\gamma}^{ab}(p) \,. \tag{39}$$

## 6. Vacuum and Discrete Analogs of Grassmann Densities

In [32], the author proposed a method for constructing a state vector of the vacuum. Let us analyze it in more detail. We divide the momentum space into infinitely small volumes. We introduce operators

$$B_k(p_j) = \frac{1}{\triangle^3 p_j} \int\limits_{\triangle^3 p_j} d^3p \, b_k(p) \,,$$

$$\bar{B}_k(p_j) = \frac{1}{\triangle^3 p_j} \int\limits_{\triangle^3 p_j} d^3p \, \bar{b}_k(p) \,. \tag{40}$$

At the same time, given Equation (34),

$$\{\bar{B}_k(p_i), B_l(p_j)\} = \frac{1}{\triangle^3 p_i \triangle^3 p_j} \int\limits_{\triangle^3 p_i} d^3p \int\limits_{\triangle^3 p_j} d^3p' \{\bar{b}_k(p), b_l(p')\} = \frac{1}{\triangle^3 p_j} \delta^i_j \delta^k_l \,. \tag{41}$$

There is no silent summation over the index that enumerates discrete volumes. For example, it does not exist at index $j$ in Equations (40) and (41). For indexes enclosed in triangular brackets (for example, in Equation (43)), there is also no silent summation.

The expression $\frac{1}{\triangle^3 p_j} \delta^i_j$ in Equations (40) and (41) is a discrete analog of the delta function $\delta(p - p')$.

In addition, due to the anticommutativity of all $b_k(p)$ and $b_l(p')$ as well as all $\bar{b}_k(p)$ and $\bar{b}_l(p')$, it is obvious that

$$(B_k(p_j))^2 = (\bar{B}_k(p_j))^2 = 0 \,. \tag{42}$$

We introduce operators

$$\Psi_{B_k j} = \triangle^3 p_j B_{<k>}(p_j) \bar{B}_{<k>}(p_j) \,,$$

$$\Psi_{V_j} = \Psi_{B_1 j} \Psi_{B_2 j} \Psi_{B_3 j} \Psi_{B_4 j} \tag{43}$$

and determine via them the fermionic vacuum operator $\Psi_V$

$$\Psi_V = \prod_j \Psi_{V_j} \,, \tag{44}$$

where the product goes over all physically possible values of $j$. In this case, we assume that all volumes $\triangle^3 p_j$ are formed by Lorentz rotations of the volume $\triangle^3 p_{j=0}$ corresponding to $p = 0$, and the set of angles $\omega_{\mu\nu}$ of these rotations is discrete.

Further, it is often convenient to represent Equation (44) in the form

$$\Psi_V = \Psi_{V_j} \Psi'_{V_j} \,, \tag{45}$$

where

$$\Psi'_{V_j} = \prod_{i \neq j} \Psi_{V_i} \,, \tag{46}$$

is the product of factors in Equation (44), independent of $p_j$.

Let us replace in the formulas with participation of $\hat{\gamma}^a$ and $\hat{\gamma}^{ab}$ continuous operators $b_k(p)$ and $\bar{b}_k(p)$ to discrete $B_k(p_j)$ and $\bar{B}_k(p_j)$, and the integral $\int d^3p \,...$ to the sum $\sum_j \triangle^3 p_j \,...$. In this case, all formulas using continuous operators $b_k(p)$ and $\bar{b}_k(p)$ are replaced by completely similar ones, with the replacement of the delta function $\delta(p - p')$ by $\frac{1}{\triangle^3 p_j} \delta^i_j$, where $p_i$ corresponds to $p$, and $p_j$ corresponds

to $p'$. We use for operators $\hat{\gamma}^a = \sum_j \triangle^3 p_j \hat{\gamma}^a(p_j)$ and $\hat{\gamma}^{ab} = \sum_j \triangle^3 p_j \hat{\gamma}^{ab}(p_j)$ after such a replacement the same notation as for the corresponding continuous ones, and we call such $\hat{\gamma}^a$ as discrete gamma operators, and $\hat{\gamma}^{ab}$ as discrete gamma operators of rotations.

## 7. Action of Gamma Operators on the Vacuum

Consider the action of the gamma operator of rotation $\hat{\gamma}^{\mu\nu}$ on the vacuum in Equation (45). The invariance of the vacuum during Lorentz rotation by means of the operator in Equation (26) is ensured by the fact that each volume $\triangle^3 p_j$ passes into another volume $\triangle^3 p_k$, and its place is occupied by the third volume $\triangle^3 p_l$. This only leads to a change in the order of the factors $\Psi_{V_j}$ in Equation (44). These factors commute, thus the Lorentz rotations leave the vacuum $\Psi_V$ invariant. At infinitesimal rotations, we have $T = exp(\hat{\gamma}^{\mu\nu} d\omega_{\mu\nu}/4) = 1 + \hat{\gamma}^{\mu\nu} d\omega_{\mu\nu}/4$, therefore

$$T\Psi_V = (1 + \hat{\gamma}^{\mu\nu} d\omega_{\mu\nu}/4)\Psi_V = \Psi_V \,, \tag{47}$$

that is

$$\hat{\gamma}^{\mu\nu} d\omega_{\mu\nu}\Psi_V = 0 \,. \tag{48}$$

Since the rotation parameters $d\omega_{\mu\nu}$ are arbitrary, we get

$$\hat{\gamma}^{\mu\nu}\Psi_V = 0 \,. \tag{49}$$

Dividing the momentum space into infinitely small volumes is a kind of regularization. Only within the framework of this regularization, both Equation (47) and Equation (49) resulting from it are fulfilled.

If axes $\hat{\gamma}^4$, $\hat{\gamma}^6$ and $\hat{\gamma}^7$ are considered as additional spacetime axes, the reasoning is similar with the same vacuum as in Equation (45). In this case, $\mu, \nu = 0, 1, 2, 3, 4, 6, 7$, and the integration is performed over the six-dimensional momentum space. If axis $\hat{\gamma}^4$ is excluded, the momentum space turns out to be five-dimensional.

As a result of the Lorentz transformation in Equation (26), the Grassmann densities $\frac{\partial}{\partial \theta^\alpha(p)}$ and $\theta^\alpha(p)$ are transformed into equivalent ones $\frac{\partial}{\partial \theta^\alpha(p')}$ and $\theta^\alpha(p')$, and arbitrary vector $\hat{\gamma}^\mu A_\mu$ is transformed into equivalent one $(\hat{\gamma}^\mu)' A_\mu$, where $(\hat{\gamma}^\mu)' = T\hat{\gamma}^\mu T^{-1}$. The transformations in Equation (32) are symmetry transformations. Therefore, Lorentz transformations are symmetry transformations of spinors and spacetime.

It should be noted that operators $\hat{\gamma}^{\mu\nu}$ make sense only within the framework of the decomposition in Equation (32). That is, they make sense only as generators of active Lorentz rotations (Equation (26)), which lead to a change in the momentum $p$.

Now, consider operators $\hat{\gamma}^\mu$. They also make sense only within the framework of the decomposition in Equation (32). As a result of the action of the transformation operator

$$T_1 = exp(i\hat{\gamma}^\mu d\omega_\mu) = 1 + i\hat{\gamma}^\mu d\omega_\mu \tag{50}$$

the Grassmann densities $\frac{\partial}{\partial \theta^\alpha(p)}$ and $\theta^\alpha(p)$ are transformed into equivalent ones. However, it is not the Lorentz rotation operator, and it is not symmetry transformation of the spacetime. Operators $\hat{\gamma}^\mu$ are basic vectors of a vector space. Therefore, we can identify parameters $d\omega_\mu$ with local coordinates of spinors in this vector space. In the simplest case, we can consider them as local coordinates $dx^\mu$ in the spacetime with spinor mass $m$ as a scale factor [33]

$$d\omega_\mu = - m \, dx_\mu \,, \tag{51}$$

since the decomposition of the field operator of the spinor in momenta coincides with the decomposition of the field operator in the secondary quantization formalism. In the general case,

the situation is more complicated, and it is necessary to take into account the existence of additional vector fields (see Section 10). However, it follows from the above that gamma operators $\hat{\gamma}^{\mu}$, when acting on $\frac{\partial}{\partial \theta^{\alpha}(p)}$ and $\theta^{\alpha}(p)$, do not change the momentum $p$ as their parameter. Therefore, when $\hat{\gamma}^{\mu}$ acts on field operators, vacuum and state vectors, the parameter $p$ remains unchanged.

With this in mind, consider the action of $\hat{\gamma}^0$ on the vacuum. Since $\hat{\gamma}^0$ is a commutator, we have

$$\hat{\gamma}^0 \Psi_V = \hat{\gamma}^0 \prod_j \Psi_{V_j} = (\hat{\gamma}^0 \Psi_{V_0}) \Psi_{V_1} \Psi_{V_2} \ldots + \Psi_{V_0}(\hat{\gamma}^0 \Psi_{V_1}) \Psi_{V_2} \ldots + \Psi_{V_0} \Psi_{V_1}(\hat{\gamma}^0 \Psi_{V_2}) \ldots + \ldots, \quad (52)$$

Here, brackets limit the scope of the commutator $\hat{\gamma}^0$. In this case, from Equation (43), it follows that

$$\Psi_{V_j} = (\triangle^3 p_j)^4 B_1(p_j)\bar{B}_1(p_j) B_2(p_j)\bar{B}_2(p_j) B_3(p_j)\bar{B}_3(p_j) B_4(p_j)\bar{B}_4(p_j). \quad (53)$$

Taking into account the introduced notation for discrete operators as well as the fact that an arbitrary spatial momentum can be obtained from the state with $p = 0$ (Equation (33)),

$$B_1(p_j) = e^{\hat{\gamma}^{0k}\omega_{0k}/2} B_1(0). \quad (54)$$

At the same time, $B_1(p_j)$ means that the result of rotation of a state with $p = 0$ turns into the state with $p = p_j$.

First, consider action of $\hat{\gamma}^0(0)$ on $\Psi_V$. Since

$$\Psi_{V_0} = \frac{\partial}{\partial \theta^1(0)} \theta^1(0) \frac{\partial}{\partial \theta^2(0)} \theta^2(0) \frac{\partial}{\partial \theta^3(0)} \theta^3(0) \frac{\partial}{\partial \theta^4(0)} \theta^4(0), \quad (55)$$

it is easy to see that

$$\hat{\gamma}^0(0)\Psi_V = (\hat{\gamma}^0(0)\Psi_{V_0}) \Psi_{V_1} \Psi_{V_2} \ldots =$$
$$- \triangle^3 p_j [\theta^k(0) \frac{\partial}{\partial \theta^k(0)}, \frac{\partial}{\partial \theta^1(0)} \theta^1(0) \frac{\partial}{\partial \theta^2(0)} \theta^2(0) \frac{\partial}{\partial \theta^3(0)} \theta^3(0) \frac{\partial}{\partial \theta^4(0)} \theta^4(0)] \Psi_{V_1} \Psi_{V_2} \ldots = 0. \quad (56)$$

Now, consider action of $\hat{\gamma}^0(p)$ on $\Psi_V$ for the case when continuous momentum $p = p1$ with corresponding discrete $p_j$, that is, it is directed along the axis $\hat{\gamma}^1$. Let us represent $\Psi_V$ as a product $\Psi_V = \Psi_{V_{1,4}} \Psi_{V_{2,3}} \Psi'_V$ where

$$\Psi_{V_{1,4}} = (\triangle^3 p_j)^2 B_1(p_j)\bar{B}_1(p_j) B_4(p_j)\bar{B}_4(p_j)$$
$$\Psi_{V_{2,3}} = (\triangle^3 p_j)^2 B_2(p_j)\bar{B}_2(p_j) B_3(p_j)\bar{B}_3(p_j). \quad (57)$$

Obviously,

$$\hat{\gamma}^0(p1)\Psi_V = ((\hat{\gamma}^0(p1)\Psi_{V_{1,4}}) \Psi_{V_{2,3}} + \Psi_{V_{1,4}}(\hat{\gamma}^0(p1)\Psi_{V_{2,3}})) \Psi'_V. \quad (58)$$

Write the following useful relationships:

$$\frac{\partial}{\partial \theta^{<a>}(p_j)} \theta^b(p_j) \frac{\partial}{\partial \theta^{<a>}(p_j)} = (\frac{1}{\triangle^3 p_j} \delta_a^b - \theta^b(p_j) \frac{\partial}{\partial \theta^{<a>}(p_j)}) \frac{\partial}{\partial \theta^{<a>}(p_j)} = \frac{1}{\triangle^3 p_j} \delta_a^b \frac{\partial}{\partial \theta^{<a>}(p_j)},$$
$$\theta^{<a>}(p_j) \frac{\partial}{\partial \theta^b(p_j)} \theta^{<a>}(p_j) = (\frac{1}{\triangle^3 p_j} \delta_a^b - \frac{\partial}{\partial \theta^b(p_j)} \theta^{<a>}(p_j)) \theta^{<a>}(p_j) = \frac{1}{\triangle^3 p_j} \delta_a^b \theta^{<a>}(p_j). \quad (59)$$

Consider action of $\hat{\gamma}^0(p1)$ on $\Psi_{V_{1,4}}$ and $\Psi_{V_{2,3}}$. From Equations (54), (A3) and (57), taking into account Equation (59), we obtain with $p_j = p1$

$$\hat{\gamma}^0(p1)\Psi_{V_{1,4}} = \frac{\triangle^3 p_j}{2} sinh\omega_{01}(p1)\left(\frac{\partial}{\partial\theta^4(p1)}\frac{\partial}{\partial\theta^1(p1)} + \theta^1(p1)\theta^4(p1)\right),$$

$$\hat{\gamma}^0(p1)\Psi_{V_{2,3}} = \frac{\triangle^3 p_j}{2} sinh\omega_{01}(p1)\left(\frac{\partial}{\partial\theta^3(p1)}\frac{\partial}{\partial\theta^2(p1)} + \theta^2(p1)\theta^3(p1)\right). \tag{60}$$

To understand the meaning of Equation (60), we consider the action of the operator of creation of a fermion–antifermion pair $\triangle^3 p\bar{B}_1(p)\bar{B}_4(p) \approx \triangle^3 p\theta^1(p1)\theta^4(p1)$ on $\Psi_{V_{1,4}}$ when $p \rightarrow 0$. The multiplier $\triangle^3 p$ is necessary for normalization to the unit probability of finding spinors in the whole space.

That is, $\hat{\gamma}^0(p1)\Psi_{V_{1,4}}$ contains a term corresponding to the creation of a fermion–antifermion pair $\theta^1(p1)\theta^4(p1)$, suppressed by a small multiplier $sinh\omega_{01}(p1)$ in the non-relativistic limit. $\hat{\gamma}^0(p1)\Psi_{V_{2,3}}$ corresponds to the creation of a pair $\theta^2(p1)\theta^3(p1)$ with different values of the spin.

Similarly, $\frac{\partial}{\partial\theta^4(p1)}\frac{\partial}{\partial\theta^1(p1)}$ are the creation operators of a fermion–antifermion pair for an alternative vacuum (see Section 3), where factors $\frac{\partial}{\partial\theta^1(p1)}\theta^1(p1)\frac{\partial}{\partial\theta^4(p1)}\theta^4(p1)$ in the vacuum state vector are replaced by $\theta^1(p1)\frac{\partial}{\partial\theta^1(p1)}\theta^4(p1)\frac{\partial}{\partial\theta^4(p1)}$, and similarly for operator $\frac{\partial}{\partial\theta^3(p1)}\frac{\partial}{\partial\theta^2(p1)}$ for a corresponding alternative vacuum.

Thus, $\hat{\gamma}^0(p1)\Psi_V \rightarrow 0$ when $p1 \rightarrow 0$.

Carrying out spatial rotations $exp(\hat{\gamma}^{kl}\omega_{kl}/4)$, where $k, l = 1, 2, 3$, of Equation (60), does not affect the multiplier $\hat{\gamma}^0$, since $\hat{\gamma}^{kl}$ commutes with $\hat{\gamma}^0$, we get a similar result for arbitrary directions of the spatial momentum. Thus, in the non-relativistic limit $p \rightarrow 0$, it can be considered that $\hat{\gamma}^0\Psi_V = 0$.

Similarly, we find the result of the action of $\hat{\gamma}^1(0)$ on the multipliers of $\Psi_V$:

$$\hat{\gamma}^1(0)\Psi_{V_{1,4}} = (\triangle^3 p_j)^3\left(\frac{\partial}{\partial\theta^4(0)}\frac{\partial}{\partial\theta^1(0)} + \theta^1(0)\theta^4(0)\right),$$

$$\hat{\gamma}^1(0)\Psi_{V_{2,3}} = (\triangle^3 p_j)^3\left(\frac{\partial}{\partial\theta^3(0)}\frac{\partial}{\partial\theta^2(0)} + \theta^2(0)\theta^3(0)\right). \tag{61}$$

That means the creation of fermion–antifermion pairs by operator $\hat{\gamma}^1$ even at zero momentum, without suppression of this process in the non-relativistic limit. In the case $p = 0$, the state vector has the form of Equation (10). Consider a single-particle state for which $\phi = \phi_\alpha \theta^\alpha(0)$. Let

$$\hat{\gamma}^1(0)\phi\Psi_V = (\hat{\gamma}^1(0)\phi)\Psi_V + \phi(\hat{\gamma}^1(0)\Psi_V) = \lambda\phi\Psi_V. \tag{62}$$

However, it is easy to verify that $(\hat{\gamma}^1(0)\phi)\Psi_V = 0$, which means

$$\phi_\alpha\theta^\alpha(0)(\hat{\gamma}^1(0)\Psi_V) = \lambda\phi\Psi_V. \tag{63}$$

If $\phi_1 \neq 0$, in accordance with Equations (61) and (55), the left side of Equation (63) contains a nonzero term with factor $\theta^1(0)\theta^2(0)\theta^3(0)$. However, in the right part, there is no such term. A similar situation is observed for all other cases ($\phi_2 \neq 0, \phi_3 \neq 0, \phi_4 \neq 0$).

Thus, operator $\hat{\gamma}^1$ in the non-relativistic limit $p \rightarrow 0$ (and, therefore, in general) cannot have eigenvalues on single-particle state vectors. This means that this operator cannot correspond to the operator of physical measurable quantity. Among other things, this means that this operator cannot correspond to the operator $\gamma^0$ of the Dirac theory of spinors, even if it is multiplied by $i$.

We get the same situation when acting on the vacuum and on state vectors by operators $\hat{\gamma}^a$, $a = 1, 2, 3, 4, 6, 7$—they do not annulate the vacuum in the non-relativistic limit and cannot have eigenvalues on single-particle state vectors.

## 8. Lorentz-Invariant Gamma Operators

It is easy to construct Lorentz-invariant analogs $\hat{\Gamma}^a$ and $\hat{\Gamma}^{ab}$ of superalgebraic representations $\hat{\gamma}^\mu$ of Dirac matrices and rotation generators $\hat{\gamma}^{\mu\nu}$. To do this, it is enough in Equations (A2) and (A3) to replace all operators $\frac{\partial}{\partial\theta^k(p)}$ by $b_k(p)$, and operators $\theta^k(p)$ by $\bar{b}_k(p)$. For example,

$$\hat{\Gamma}^0 = \int d^3p\,[b_1(p)\bar{b}_1(p) + b_2(p)\bar{b}_2(p) + b_3(p)\bar{b}_3(p) + b_4(p)\bar{b}_4(p), *]\,, \tag{64}$$

$$\hat{\Gamma}^1 = \int d^3p\,[b_1(p)b_4(p) - \bar{b}_4(p)\bar{b}_1(p) + b_2(p)b_3(p) - \bar{b}_3(p)\bar{b}_2(p), *]\,, \tag{65}$$

$$\hat{\Gamma}^{67} = -i\int d^3p\,[b_1(p)\bar{b}_1(p) + b_2(p)\bar{b}_2(p) - b_3(p)\bar{b}_3(p) - b_4(p)\bar{b}_4(p), *]\,, \tag{66}$$

and so on (see Equations (A4) and (A5)).

In the discrete version of the theory, in the operators $\hat{\Gamma}^a$ and $\hat{\Gamma}^{ab}$, as above, continuous operators $b_k(p)$ and $\bar{b}_k(p)$ are replaced by discrete $B_k(p_j)$ and $\bar{B}_k(p_j)$, and integrals $\int d^3p\,...$ by sums $\sum_j \triangle^3 p_j\,...$.

Operators $\hat{\Gamma}^\mu$ and $\hat{\Gamma}^{\mu\nu}$ are constructed by summing (integrating in the continuous case) over spatial momenta the results of all possible Lorentz rotations of operators $\hat{\gamma}^\mu(0)$ and $\hat{\gamma}^{\mu\nu}(0)$. As a result of such rotations, $\frac{\partial}{\partial\theta^k(0)}$ goes to $b_k(p)$, and $\theta^k(0)$ to $\bar{b}_k(p)$ as in the field operators, as in $\hat{\gamma}^\mu(0)$ and $\hat{\gamma}^{\mu\nu}(0)$.

In contrast to $\hat{\gamma}^\mu$ and $\hat{\gamma}^{\mu\nu}$, operators $\hat{\Gamma}^a$ and $\hat{\Gamma}^{ab}$ do not change either in the Lorentz transformations, since, as for the vacuum, the sum element for some momentum goes into the sum element for another momentum, and the sum element for the third momentum takes its place. As a result, these operators are Lorentz-invariant (and therefore also Lorentz-covariant). For the same reason, if for some values of $\mu$ and $\nu$ operator $\hat{\gamma}^\mu(0)$ or $\hat{\gamma}^{\mu\nu}(0)$ annulate the vacuum, then $\hat{\Gamma}^\mu$ or $\hat{\Gamma}^{\mu\nu}$ annulate the vacuum too, and if $\hat{\gamma}^\mu(0)$ or $\hat{\gamma}^{\mu\nu}(0)$ do not annulate the vacuum, then $\hat{\Gamma}^\mu$ or $\hat{\Gamma}^{\mu\nu}$ under the action on the vacuum do not give zero. For the same reason, if $\hat{\gamma}^\mu(0)$ or $\hat{\gamma}^{\mu\nu}(0)$ has eigenvalue for the state with $p = 0$, then $\hat{\Gamma}^\mu$ or $\hat{\Gamma}^{\mu\nu}$ has corresponding eigenvalue for states with any momenta. That is why operators $\hat{\Gamma}^\mu$ have the same signature as $\hat{\gamma}^\mu(0)$ and, hence, the same signature as $\hat{\gamma}^\mu$.

Operators $\hat{\gamma}^{\mu\nu}(0)$ annulate the vacuum. Operator $\hat{\gamma}^0$ annulate the vacuum only in the non-relativistic limit $p \to 0$. Operators $\hat{\gamma}^k$, $k = 1, 2, 3, 4, 6, 7$, do not annulate the vacuum. Therefore, in quantum relativistic field theory, eigenvalues of operators $\hat{\gamma}^{\mu\nu}$, $\hat{\Gamma}^0$ and $\hat{\Gamma}^{ab}$ exist on the state vectors, and operators $\hat{\gamma}^\mu$ cannot have eigenvalues at all, since they do not annulate the vacuum.

Since the commutation relations in Equation (34) for $b_i(p)$ and $\bar{b}_k(p)$ are the same as for $\frac{\partial}{\partial\theta^i(p)}$ and $\theta^k(p)$, the commutation relations for $\hat{\Gamma}^a$ and $\hat{\Gamma}^{ab}$ are the same as for $\hat{\gamma}^\mu$ and $\hat{\gamma}^{\mu\nu}$. That is, $\hat{\Gamma}^a$, $a = 0, 1, 2, 3, 4$ are also analogs of Dirac matrices $\gamma^\mu$, $\mu = 0, 1, 2, 3, 4$, but $\hat{\Gamma}^6$ and $\hat{\Gamma}^7$ also expand the set of analogs of Dirac matrices as $\hat{\gamma}^6$ and $\hat{\gamma}^7$.

We introduce the superalgebraic analogs [32] of the operators of the number of particles $\hat{N}_1$, $\hat{N}_2$ and antiparticles $\hat{N}_3$, $\hat{N}_4$ and the charge operator $\hat{Q}$ in the method of second quantization:

$$\hat{N}_k(p) = [\bar{b}_{<k>}(p)\,b_{<k>}(p), *] = -[b_{<k>}(p)\,\bar{b}_{<k>}(p), *]\,,$$
$$\hat{Q} = \int d^3p\,(\hat{N}_1(p) + \hat{N}_2(p) - \hat{N}_3(p) - \hat{N}_4(p))\,, \tag{67}$$

Then, the physical meaning of $\hat{\Gamma}^0$ and $\hat{\Gamma}^{67}$ is obvious, since Equations (64) and (66) can be rewritten in the form:

$$\hat{\Gamma}^0 = -\int d^3p\,(\hat{N}_1(p) + \hat{N}_2(p) + \hat{N}_3(p) + \hat{N}_4(p))\,,$$
$$\hat{\Gamma}^{67} = i\int d^3p\,(\hat{N}_1(p) + \hat{N}_2(p) - \hat{N}_3(p) - \hat{N}_4(p)) = i\hat{Q}\,. \tag{68}$$

That is, $-\hat{\Gamma}^0$ is the operator of the total number of spinors and antispinors, and $\hat{\Gamma}^{67}$ is related to the charge operator $\hat{Q}$ by the relation $\hat{\Gamma}^{67} = i\hat{Q}$. However, the physical meaning of operators $\hat{\Gamma}^k$ and $\hat{\Gamma}^{ab}$, where $k = 1, 2, 3, 4, 6, 7$; $a, b = 0, 1, 2, 3, 4, 6, 7$; $a \neq b$; $ab \neq 67$, is incomprehensible.

It is useful to note that the matrix formalism does not provide the possibility of zero eigenvalues of gamma matrices, in contrast to the proposed theory.

## 9. Spacetime Signature in the Presence of the Spinor Vacuum

The reason for the difference between the action on the vacuum and the state vectors of the operators $\hat{\gamma}^0(0)$ on the one hand, and $\hat{\gamma}^k(0)$, $k = 1, 2, 3, 4, 6, 7$, on the other hand, is related to the structure of these operators in Equation (A2). Since the vacuum state vector has a multiplier $B_{<i>}(0)\,\bar{B}_{<i>}(0)$, the action on vacuum of operators consisting only of terms of the form $[\bar{B}_l(0)B_r(0), *]$ will always give zero, since, by virtue of Equations (41) and (42),

$$[\bar{B}_{<l>}(0)B_{<r>}(0),\ B_{<r>}(0)\bar{B}_{<r>}(0)B_{<l>}(0)\bar{B}_{<l>}(0)] = 0. \tag{69}$$

However, the terms of the form $[B_r(0)\,B_l(0), *]$ and $[\bar{B}_r(0)\,\bar{B}_l(0), *]$ will give a non-zero result. Summing the results of Lorentz rotations leads to similar conclusions for $\hat{\Gamma}^0$ on the one hand, and $\hat{\Gamma}^k$ on the other hand.

The decomposition in Equation (32) generates the decomposition of field operators with respect to momenta and leads to the Dirac equation [33]. The question arises of what kind of Clifford basis such decomposition is possible.

If, as in the considered case, $\hat{\gamma}^0 = (\hat{\gamma}^0)^+$, $\hat{\gamma}^k = -(\hat{\gamma}^k)^+$, there is one time-like Clifford vector.

Multiplying $\hat{\gamma}^0$ by an imaginary unit will lead to the appearance in the decomposition with respect to momenta [33] of exponentially increasing terms, that is, to the impossibility of the existence of normalized solutions. Therefore, Clifford vectors $\hat{\gamma}^0$ and $\hat{\Gamma}^0$ are time-like and have signature +1 for spacetime where spinors can exist as physical particles.

Multiplication of any of operators $\hat{\gamma}^k$ (and, consequently, $\hat{\Gamma}^k$) by the imaginary unit will lead to asymmetry between Clifford vectors $\hat{\Gamma}^0$ and $i\hat{\Gamma}^k$ due to the presence of the vacuum in Equation (44), since $\hat{\Gamma}^0\Psi_V = 0$ and $i\hat{\Gamma}^k\Psi_V \neq 0$, and $\hat{\Gamma}^0$ can have eigenvalues on the state vectors but $i\hat{\Gamma}^k$ cannot. The space of Clifford vectors with the same signature must be isotropic, however in this case we obtain a preferred direction. Therefore, other than $\hat{\Gamma}^0$ Clifford vectors could not have the same signature as $\hat{\Gamma}^0$. Consequently, the condition for the existence of the vacuum imposes restrictions on the possible variants of Clifford algebras: neither complex algebra nor algebras in which at least one of the base vectors $\hat{\Gamma}^k$ (and hence $\hat{\gamma}^k$) is time-like is suitable. Therefore, all Clifford vectors $\hat{\Gamma}^k$ are space-like (and hence $\hat{\gamma}^k$)—they have a signature of $-1$, and there is only one basis time-like Clifford vector $\hat{\Gamma}^0$ (and hence $\hat{\gamma}^0$).

Operator $\hat{\Gamma}^0$ annulate the vacuum, and $\hat{\Gamma}^k, k = 1, 2, 3, 4, 6, 7$, do not annulate. Therefore, if we require the existence of spinors as physical particles with second quantization of spinor fields and existence of state vectors, out of seven gamma matrices $\hat{\Gamma}^a$ (and hence $\hat{\gamma}^a$), one must have a positive signature, and the other six must have a negative signature.

Thus, in the superalgebraic theory of spinors, the signature of a four-dimensional spacetime can only be $(1, -1, -1, -1)$, and there are two additional axes $\hat{\gamma}^6$ and $\hat{\gamma}^7$ with a signature $(-1, -1)$ corresponding to the inner space of the spinor. The reason they and the axis $\hat{\gamma}^4$ are not additional spatial axes is not yet clear.

Of course, the conclusions made about the spacetime signature rely on some assumptions. First, it is assumed that the proposed formalism is consistent with the principles of measurement of physical quantities in quantum mechanics. Secondly, it is assumed that the spinor vacuum as well as space of Clifford vectors with the same signature are isotropic.

## 10. Internal Degrees of Freedom of Superalgebraic Spinors

In [33], it was shown that transformations of densities $\theta^a(p)$ and $\frac{\partial}{\partial \theta^a(p)}$, while maintaining their CAR-algebra of creation and annihilation operators, provide the transformations in Equation (32) of field operators.

If we replace $\theta^a(p)$ and $\frac{\partial}{\partial \theta^a(p)}$ in the method proposed in [33] with $\bar{b}(p)$ and $b(p)$, we get a decomposition similar to Equation (32)

$$\Psi(p)' = (1 + i\hat{\Gamma}^a d\omega_a + \frac{1}{4}\hat{\Gamma}^{ab} d\omega_{ab})\Psi(p),\tag{70}$$

where $a, b = 0, 1, 2, 3, 4, 6, 7$ and $d\omega_{ab} = -d\omega_{ba}$ are real infinitesimal transformation parameters.

This decomposition occurs due to the presence of internal degrees of freedom of superalgebraic spinors. Consider the decomposition in Equation (70) in the case of an infinitely small change in coordinates $dx^\mu$. We have $d\omega_a = A_{a\mu}dx^\mu$ and $d\omega_{ab} = A_{ab\mu}dx^\mu$. That is why

$$\Psi(p)' = (1 + i\hat{\Gamma}^a A_{a\mu}dx^\mu + \frac{1}{4}\hat{\Gamma}^{ab} A_{ab\mu}dx^\mu)\Psi(p),\tag{71}$$

or equivalently

$$d\Psi(p) = \Psi(p)' - \Psi(p) = (i\hat{\Gamma}^a A_{a\mu}dx^\mu + \frac{1}{4}\hat{\Gamma}^{ab} A_{ab\mu}dx^\mu)\Psi(p).\tag{72}$$

We denote $A_{0\mu} = -p_\mu$ and assume that the parameter $p$ in Equations (71) and (72) refers to $p_\mu$. Consider in Equation (72) the term corresponding to $p_\mu$

$$d_0\Psi(p) = -i\hat{\Gamma}^0 p_\mu dx^\mu \Psi(p) = -i\int d^3p' \,\hat{\Gamma}^0(p')\, p_\mu dx^\mu \Psi(p) = -i\int d^3p'\, p'_\mu\, \hat{\Gamma}^0(p')\, dx^\mu \Psi(p).\tag{73}$$

Since

$$\hat{P}_\mu = \int d^3p\, p_\mu\, \hat{\Gamma}^0(p) = \int d^3p\, p_\mu\, [b_1(p)\bar{b}_1(p) + b_2(p)\bar{b}_2(p) + b_3(p)\bar{b}_3(p) + b_4(p)\bar{b}_4(p), *]\tag{74}$$

is the operator of energy–momentum in the second quantization formalism, we have

$$d_0\Psi(p) = -i\hat{P}_\mu dx^\mu \Psi(p).\tag{75}$$

The superposition of all possible values of the momentum gives

$$\Psi = \int d^3p\, \Psi(p),\tag{76}$$

and we have

$$d_0\Psi = -i\hat{P}_\mu dx^\mu \Psi.\tag{77}$$

Thus, term $i\hat{\Gamma}^0 A_{0\mu}dx^\mu = -i\hat{\Gamma}^0 p_\mu dx^\mu$ corresponds to the decomposition with respect to momenta, and our assumption that the parameter $p$ in Equations (71) and (72) refers to $p_\mu$ is proper. If we have more than three spatial axes, all the calculations are similar, only the integration over the momentum will be carried out not over three, but over all spatial components.

Values $A_{a\mu}$ and $A_{ab\mu}$ are affine connections. Thus, it is possible to build a theory of Clifford and spinor bundles without restrictions imposed by the requirement of covariantly constant idempotent field.

Now, consider the term

$$d_{67}\Psi = \frac{1}{2}\hat{\Gamma}^{67} A_{67\mu}dx^\mu \Psi.\tag{78}$$

Let $\hat{Q} = -i\hat{\Gamma}^{67}$ and $A_{67\mu} = g\,A_\mu$, where $A_\mu$ is the vector field and $g$ is the coupling constant for this field. Then,

$$d_{67}\Psi = i\frac{1}{2}g\hat{Q}A_\mu dx^\mu\Psi\,, \tag{79}$$

and the gauge transformation $exp(i\frac{1}{2}g\hat{Q}A_\mu dx^\mu)$ automatically arises. Operator $\hat{Q}$ is the charge operator in the second quantization formalism, $\hat{Q}\Psi = \Psi$ for the spinor $\Psi$, and $\hat{Q}\bar{\Psi} = -\bar{\Psi}$ for its antiparticle $\bar{\Psi}$.

It should be noted that in Equation (77) we can replace $-i\hat{P}_\mu$ with $\partial_\mu$. In accordance with Equation (72), taking into account Equations (77) and (79), we can write the covariant derivative in the form

$$D_\mu = \partial_\mu + i\frac{1}{2}g\hat{Q}A_\mu + i\hat{\Gamma}^a A_{a\mu} + \frac{1}{4}\hat{\Gamma}^{bc}A_{bc\mu}\,, \tag{80}$$

where $a = 1,2,3,4,6,7$; $b,c = 0,1,2,3,4,6,7$ and $bc \neq 67$, $bc \neq 76$.

Part of decomposition terms in Equation (80) corresponds to the usual field theories available in the framework of the general theory of relativity [34], as well as to theories of bundles [35]. The physical meaning of the other terms requires additional research. In any case, the proposed approach opens perspectives for constructing a theory in which properties of the spacetime have the same algebraic nature as the momentum, electromagnetic field and other quantum fields.

## 11. Discussion

Thus, the superalgebraic formulation of the theory of algebraic spinors allows constructing composite analogs of the Dirac gamma matrices. The proposed theory has a number of interesting consequences.

Equations (32) and (70) ensure for spinors the existence of the decomposition with respect to momenta.

The theory is free from divergences, leading to the need for the normalization of operators [32].

It leads to an unambiguous signature of the spacetime, which coincides with the observable.

The proposed approach of constructing a discrete vacuum is fundamentally different from theories in which the discreteness of the spacetime is considered, leading to the loss of Lorentz covariance [36]. The proposed theory is Lorentz-covariant and combines the features of discrete and continuous theories.

We can construct $\hat{\Gamma}^a$ of creation and annihilation operators independently on superalgebraic representation $\hat{\gamma}^\mu$ of Dirac gamma matrices $\gamma^\mu$. However, this representation makes interconnection between Dirac gamma matrices and operators $\hat{\Gamma}^a$ obvious.

At the same time, there are several unsolved problems in the proposed theory.

First, gamma operators $\hat{\gamma}^4$, $\hat{\gamma}^6$ and $\hat{\gamma}^7$ (and, respectively, $\hat{\Gamma}^4$, $\hat{\Gamma}^6$ and $\hat{\Gamma}^7$) have exactly the same properties as operators $\hat{\gamma}^1$, $\hat{\gamma}^2$ and $\hat{\gamma}^3$ (and $\hat{\Gamma}^1$ and $\hat{\Gamma}^2$, $\hat{\Gamma}^3$, respectively). Therefore, it seems that the spatial dimensions should be six, not three. However, the equality holds in the small Clifford algebra

$$\hat{\gamma}^0\hat{\gamma}^1\hat{\gamma}^2\hat{\gamma}^3\hat{\gamma}^4\hat{\gamma}^6\hat{\gamma}^7 = -i\,E\,, \tag{81}$$

where $E$ is the identity operator in the small Clifford algebra. Therefore, it seems that one of the gamma operators should be regarded as dependent on the others. It is natural to use $\hat{\gamma}^4$ as such operator to obtain the ordinary Dirac theory, extended by $\hat{\gamma}^6$ and $\hat{\gamma}^7$ operators. However, Equation (81) is valid only in the small Clifford algebra. Accordingly, $E$ is the identity operator only in the small Clifford algebra, since it follows from Equations (49) and (81) that $E\,\Psi_V = 0$. In addition, all gamma operators are involved in the decompositions in Equations (32) and (70). Therefore, the question of what limitations are imposed by Equation (81) requires additional study.

The reason that $\hat{\gamma}^6$ and $\hat{\gamma}^7$ axes are not the generators of the physical spatial axes may be due to the fact that operators $\hat{\gamma}^6$, $\hat{\gamma}^7$, $\hat{\gamma}^{6k}$ and $\hat{\gamma}^{7k}$, where $k = 0,1,2,3,4$, mix components of spinors and antispinors. If such mixing is prohibited, and the fermion field operators are required to be the eigenvectors of the charge operator $\hat{Q}$, then $\hat{\gamma}^6$ and $\hat{\gamma}^7$ axes can only correspond to the internal degrees

of freedom of the fermions. In this case, the fermions themselves can be either only spinors or only antispinors, but not their mixture.

Another approach is possible to solve this problem. We can consider the creation and annihilation operators, of which the gamma operators $\Gamma^a$ are constructed as primary. In addition, their properties are determined by the properties of the physical spacetime. In this case, no symmetry violation can occur between the generators of Clifford algebra having the same signature. In this case, the number of the basis Clifford vectors corresponding to the spacetime is specified by the properties of the physical spacetime. The consideration given in the article shows that the vacuum state vector and the creation and annihilation operators of spinors are possible only if the physical space-time signature is $(+1, -1, -1, -1)$. At the same time, there are internal degrees of freedom associated with the existence of gamma operators $\hat{\gamma}^6$ and $\hat{\gamma}^7$. These generators have the signature $(-1, -1)$ due to the requirement of the isotropy of the Clifford algebra vector space.

The second problem, which has not yet been solved, is related to the presence of vector fields in Equations (70) and (80), the physical meaning of which is still unclear. To solve this problem, further research is required.

Increasing the number of independent spinor densities $\theta^a(p)$ and $\frac{\partial}{\partial\theta^a(p)}$ increases the number of dimensions of small Clifford algebra. In this case, additional fields generated by affine connections appear in the decomposition in Equation (71). This allows us to hope for the construction on this basis of a theory describing all known spinors and their interactions.

## 12. Conclusions

A new formalism involving spinors in theories of space-time and vacuum is presented. It is based on a superalgebraic formulation of the theory of algebraic spinors. It is proved that the signature of four-dimensional spacetime, in which the vacuum state exists, can only be $(1, -1, -1, -1)$, and there are two additional axes corresponding to the inner space of the spinor, with a signature $(-1, -1)$.

Section 1 of the article contains information on various approaches used to determine the possible dimensions and signature of the spacetime.

Section 2 briefly describes the approaches used in the theory of algebraic spinors.

Section 3 describes the reformulation of the theory of algebraic spinors in terms of Grassmann variables and derivatives with respect to them. Hermitian primitive idempotents constructed of these variables and derivatives.

Section 4 introduces the concept of state vector. The transformations of state vectors and operators are considered. The Lie group corresponding to the Clifford group is investigated. It is shown that, in addition to the Clifford algebra under consideration, which we call the large Clifford algebra, there can be another Clifford algebra. We call it small Clifford algebra. It is shown that, in the four-dimensional case, there are seven generators of the small Clifford algebra. Five of them correspond to the Dirac matrices and two additional ones are related to the internal degrees of freedom of the spinor. Elements of the small Clifford algebra are operators. They satisfy the relations of Clifford algebra only under the action on vectors of large Clifford algebra, otherwise it is necessary to consider more complex relations.

Section 5 contains information on the construction of analogs $\hat{\gamma}^\mu$ of Dirac matrices using variables, which are Grassmann density, and derivatives with respect to them. In addition, based on the Grassmann densities and derivatives with respect to them, operators of pseudo-orthogonal rotations are constructed. These include the Lorentz transformations.

In Section 6, the transition is carried out in the momentum space from continuous Grassmann densities to infinitesimal discrete volumes. The vacuum state vector is constructed as a product of local vacua related to these discrete volumes.

Section 7 is the most important in the article. It proves that the operator $\hat{\gamma}^0$, acting on the vacuum state vector, gives zero in the non-relativistic limit $p \to 0$. Operators $\hat{\gamma}^k, k = 1, 2, 3, 4, 6, 7$, acting on

the vacuum state vector, do not give zero in the non-relativistic limit $p \to 0$. It is shown that for this reason operators $\hat{\gamma}^k$ cannot have eigenvalues on state vectors.

In Section 8, other superalgebraic analogs $\hat{\Gamma}^a$ of Dirac gamma matrices are constructed. These operators are constructed of creation and annihilation operators and have the same signature as gamma operators $\hat{\gamma}^\mu$. However, they are Lorentz-invariant. We proved that operator $-\hat{\Gamma}^0$ is the operator of the total number of spinors and antispinors, and $\hat{\Gamma}^{67}$ is related to the charge operator $\hat{Q}$ by the relation $\hat{\Gamma}^{67} = i\hat{Q}$.

In Section 9, we draw conclusions about a possible spacetime signature.

Section 10 shows that the developed mathematical formalism allows one to obtain the second quantization operators in a natural way. When the spinor coordinate changes, gauge transformations arise due to existence of internal degrees of freedom of the superalgebraic spinors. These degrees of freedom lead to existence of affine connections. The proposed approach opens perspectives for constructing a theory in which the properties of the spacetime have the same algebraic nature as the momentum, electromagnetic field and other quantum fields.

In Section 11, we discuss the advantages and possible directions for the development of the proposed approach.

**Funding:** This research received no external funding.

**Conflicts of Interest:** The author declares no conflict of interest.

**Abbreviations**

The following abbreviations are used in this manuscript:

CAR   Canonical Anticommutation Relations

**Appendix A. Formulas of Gamma Operators**

Formulas of gamma operators $\hat{\gamma}_p^a$ are as follows:

$$
\begin{aligned}
\hat{\gamma}_p^0 &= [\frac{\partial}{\partial\theta^1}\theta^1 + \frac{\partial}{\partial\theta^2}\theta^2 + \frac{\partial}{\partial\theta^3}\theta^3 + \frac{\partial}{\partial\theta^4}\theta^4, *], \\
\hat{\gamma}_p^1 &= [\frac{\partial}{\partial\theta^1}\frac{\partial}{\partial\theta^4} - \theta^4\theta^1 + \frac{\partial}{\partial\theta^2}\frac{\partial}{\partial\theta^3} - \theta^3\theta^2, *], \\
\hat{\gamma}_p^2 &= -i[\frac{\partial}{\partial\theta^1}\frac{\partial}{\partial\theta^4} - \theta^4\theta^1 + \frac{\partial}{\partial\theta^2}\frac{\partial}{\partial\theta^3} + \theta^3\theta^2, *], \\
\hat{\gamma}_p^3 &= [\frac{\partial}{\partial\theta^1}\frac{\partial}{\partial\theta^3} - \theta^3\theta^1 - \frac{\partial}{\partial\theta^2}\frac{\partial}{\partial\theta^4} + \theta^4\theta^2, *], \\
\hat{\gamma}_p^4 &= i\hat{\gamma}_p^5 = i[\frac{\partial}{\partial\theta^1}\frac{\partial}{\partial\theta^3} + \theta^3\theta^1 + \frac{\partial}{\partial\theta^2}\frac{\partial}{\partial\theta^4} + \theta^4\theta^2, *], \\
\hat{\gamma}_p^6 &= i[\frac{\partial}{\partial\theta^1}\frac{\partial}{\partial\theta^2} + \theta^2\theta^1 - \frac{\partial}{\partial\theta^3}\frac{\partial}{\partial\theta^4} - \theta^4\theta^3, *], \\
\hat{\gamma}_p^7 &= [\frac{\partial}{\partial\theta^1}\frac{\partial}{\partial\theta^2} - \theta^2\theta^1 + \frac{\partial}{\partial\theta^3}\frac{\partial}{\partial\theta^4} - \theta^4\theta^3, *].
\end{aligned}
\tag{A1}
$$

Formulas of gamma operators $\hat{\gamma}^a$ are as follows:

$$\hat{\gamma}^0 = \int d^3p \left[ \frac{\partial}{\partial \theta^1(p)} \theta^1(p) + \frac{\partial}{\partial \theta^2(p)} \theta^2(p) + \frac{\partial}{\partial \theta^3(p)} \theta^3(p) + \frac{\partial}{\partial \theta^4(p)} \theta^4(p), * \right],$$

$$\hat{\gamma}^1 = \int d^3p \left[ \frac{\partial}{\partial \theta^1(p)} \frac{\partial}{\partial \theta^4(p)} - \theta^4(p)\theta^1(p) + \frac{\partial}{\partial \theta^2(p)} \frac{\partial}{\partial \theta^3(p)} - \theta^3(p)\theta^2(p), * \right],$$

$$\hat{\gamma}^2 = i \int d^3p \left[ -\frac{\partial}{\partial \theta^1(p)} \frac{\partial}{\partial \theta^4(p)} - \theta^4(p)\theta^1(p) + \frac{\partial}{\partial \theta^2(p)} \frac{\partial}{\partial \theta^3(p)} + \theta^3(p)\theta^2(p), * \right],$$

$$\hat{\gamma}^3 = \int d^3p \left[ \frac{\partial}{\partial \theta^1(p)} \frac{\partial}{\partial \theta^3(p)} - \theta^3(p)\theta^1(p) - \frac{\partial}{\partial \theta^2(p)} \frac{\partial}{\partial \theta^4(p)} + \theta^4(p)\theta^2(p), * \right], \tag{A2}$$

$$\hat{\gamma}^4 = i\hat{\gamma}^5 = i \int d^3p \left[ \frac{\partial}{\partial \theta^1(p)} \frac{\partial}{\partial \theta^3(p)} + \theta^3(p)\theta^1(p) + \frac{\partial}{\partial \theta^2(p)} \frac{\partial}{\partial \theta^4(p)} + \theta^4(p)\theta^2(p), * \right],$$

$$\hat{\gamma}^6 = i \int d^3p \left[ \frac{\partial}{\partial \theta^1(p)} \frac{\partial}{\partial \theta^2(p)} + \theta^2(p)\theta^1(p) - \frac{\partial}{\partial \theta^3(p)} \frac{\partial}{\partial \theta^4(p)} - \theta^4(p)\theta^3(p), * \right],$$

$$\hat{\gamma}^7 = \int d^3p \left[ \frac{\partial}{\partial \theta^1(p)} \frac{\partial}{\partial \theta^2(p)} - \theta^2(p)\theta^1(p) + \frac{\partial}{\partial \theta^3(p)} \frac{\partial}{\partial \theta^4(p)} - \theta^4(p)\theta^3(p), * \right].$$

Formulas of gamma operators of rotations $\hat{\gamma}^{ab}$ are as follows:

$$\hat{\gamma}^{01} = \int d^3p \left[ \frac{\partial}{\partial \theta^1(p)} \frac{\partial}{\partial \theta^4(p)} + \theta^4(p)\theta^1(p) + \frac{\partial}{\partial \theta^2(p)} \frac{\partial}{\partial \theta^3(p)} + \theta^3(p)\theta^2(p), * \right],$$

$$\hat{\gamma}^{02} = -i \int d^3p \left[ \frac{\partial}{\partial \theta^1(p)} \frac{\partial}{\partial \theta^4(p)} - \theta^4(p)\theta^1(p) - \frac{\partial}{\partial \theta^2(p)} \frac{\partial}{\partial \theta^3(p)} + \theta^3(p)\theta^2(p), * \right],$$

$$\hat{\gamma}^{03} = \int d^3p \left[ \frac{\partial}{\partial \theta^1(p)} \frac{\partial}{\partial \theta^3(p)} + \theta^3(p)\theta^1(p) - \frac{\partial}{\partial \theta^2(p)} \frac{\partial}{\partial \theta^4(p)} - \theta^4(p)\theta^2(p), * \right],$$

$$\hat{\gamma}^{04} = i \int d^3p \left[ \frac{\partial}{\partial \theta^1(p)} \frac{\partial}{\partial \theta^3(p)} - \theta^3(p)\theta^1(p) + \frac{\partial}{\partial \theta^2(p)} \frac{\partial}{\partial \theta^4(p)} - \theta^4(p)\theta^2(p), * \right],$$

$$\hat{\gamma}^{06} = i \int d^3p \left[ \frac{\partial}{\partial \theta^1(p)} \frac{\partial}{\partial \theta^2(p)} + \theta^2(p)\theta^1(p) - \frac{\partial}{\partial \theta^3(p)} \frac{\partial}{\partial \theta^4(p)} + \theta^4(p)\theta^3(p), * \right],$$

$$\hat{\gamma}^{07} = \int d^3p \left[ \frac{\partial}{\partial \theta^1(p)} \frac{\partial}{\partial \theta^2(p)} + \theta(p)\theta^1(p) + \frac{\partial}{\partial \theta^3(p)} \frac{\partial}{\partial \theta^4(p)} + \theta^4(p)\theta^3(p), * \right],$$

$$\hat{\gamma}^{12} = -i \int d^3p \left[ \frac{\partial}{\partial \theta^1(p)} \theta^1(p) - \frac{\partial}{\partial \theta^2(p)} \theta^2(p) - \frac{\partial}{\partial \theta^3(p)} \theta^3(p) + \frac{\partial}{\partial \theta^4(p)} \theta^4(p), * \right],$$

$$\hat{\gamma}^{13} = \int d^3p \left[ \frac{\partial}{\partial \theta^1(p)} \theta^2(p) - \frac{\partial}{\partial \theta^2(p)} \theta^1(p) + \frac{\partial}{\partial \theta^3(p)} \theta^4(p) - \frac{\partial}{\partial \theta^4(p)} \theta^3(p), * \right],$$

$$\hat{\gamma}^{14} = i \int d^3p \left[ \frac{\partial}{\partial \theta^1(p)} \theta^2(p) + \frac{\partial}{\partial \theta^2(p)} \theta^1(p) + \frac{\partial}{\partial \theta^3(p)} \theta^4(p) + \frac{\partial}{\partial \theta^4(p)} \theta^3(p), * \right],$$

$$\hat{\gamma}^{16} = i \int d^3p \left[ -\frac{\partial}{\partial \theta^3(p)} \theta^1(p) - \frac{\partial}{\partial \theta^1(p)} \theta^3(p) + \frac{\partial}{\partial \theta^4(p)} \theta^2(p) + \frac{\partial}{\partial \theta^2(p)} \theta^4(p), * \right],$$

$$\hat{\gamma}^{17} = \int d^3p \left[ \frac{\partial}{\partial \theta^3(p)} \theta^1(p) - \frac{\partial}{\partial \theta^1(p)} \theta^3(p) - \frac{\partial}{\partial \theta^4(p)} \theta^2(p) + \frac{\partial}{\partial \theta^2(p)} \theta^4(p), * \right], \tag{A3}$$

$$\hat{\gamma}^{23} = -i \int d^3p \left[ \frac{\partial}{\partial \theta^1(p)} \theta^2(p) + \frac{\partial}{\partial \theta^2(p)} \theta^1(p) - \frac{\partial}{\partial \theta^3(p)} \theta^4(p) - \frac{\partial}{\partial \theta^4(p)} \theta^3(p), * \right],$$

$$\hat{\gamma}^{24} = \int d^3p \left[ \frac{\partial}{\partial \theta^1(p)} \theta^2(p) - \frac{\partial}{\partial \theta^2(p)} \theta^1(p) - \frac{\partial}{\partial \theta^3(p)} \theta^4(p) + \frac{\partial}{\partial \theta^4(p)} \theta^3(p), * \right],$$

$$\hat{\gamma}^{26} = \int d^3p \left[ \frac{\partial}{\partial \theta^3(p)} \theta^1(p) - \frac{\partial}{\partial \theta^1(p)} \theta^3(p) + \frac{\partial}{\partial \theta^4(p)} \theta^2(p) - \frac{\partial}{\partial \theta^2(p)} \theta^4(p), * \right],$$

$$\hat{\gamma}^{27} = i \int d^3p \left[ \frac{\partial}{\partial \theta^3(p)} \theta^1(p) + \frac{\partial}{\partial \theta^1(p)} \theta^3(p) + \frac{\partial}{\partial \theta^4(p)} \theta^2(p) + \frac{\partial}{\partial \theta^2(p)} \theta^4(p), * \right],$$

$$\hat{\gamma}^{34} = i \int d^3p \left[ \frac{\partial}{\partial \theta^1(p)} \theta^1(p) - \frac{\partial}{\partial \theta^2(p)} \theta^2(p) + \frac{\partial}{\partial \theta^3(p)} \theta^3(p) - \frac{\partial}{\partial \theta^4(p)} \theta^4(p), * \right],$$

$$\hat{\gamma}^{36} = i \int d^3p \left[ \frac{\partial}{\partial \theta^4(p)} \theta^1(p) + \frac{\partial}{\partial \theta^1(p)} \theta^4(p) + \frac{\partial}{\partial \theta^3(p)} \theta^2(p) + \frac{\partial}{\partial \theta^2(p)} \theta^3(p), * \right],$$

$$\hat{\gamma}^{37} = \int d^3p \left[ -\frac{\partial}{\partial \theta^4(p)} \theta^1(p) + \frac{\partial}{\partial \theta^1(p)} \theta^4(p) - \frac{\partial}{\partial \theta^3(p)} \theta^2(p) + \frac{\partial}{\partial \theta^2(p)} \theta^3(p), * \right],$$

$$\hat{\gamma}^{46} = \int d^3p \left[ \frac{\partial}{\partial \theta^4(p)} \theta^1(p) - \frac{\partial}{\partial \theta^1(p)} \theta^4(p) - \frac{\partial}{\partial \theta^3(p)} \theta^2(p) + \frac{\partial}{\partial \theta^2(p)} \theta^3(p), * \right],$$

$$\hat{\gamma}^{47} = i \int d^3p \left[ -\frac{\partial}{\partial \theta^4(p)} \theta^1(p) - \frac{\partial}{\partial \theta^1(p)} \theta^4(p) + \frac{\partial}{\partial \theta^3(p)} \theta^2(p) + \frac{\partial}{\partial \theta^2(p)} \theta^3(p), * \right],$$

$$\hat{\gamma}^{67} = -i \int d^3p \left[ \frac{\partial}{\partial \theta^1(p)} \theta^1(p) + \frac{\partial}{\partial \theta^2(p)} \theta^2(p) - \frac{\partial}{\partial \theta^3(p)} \theta^3(p) - \frac{\partial}{\partial \theta^4(p)} \theta^4(p), * \right].$$

Formulas of Lorentz invariant gamma operators $\hat{\Gamma}^a$ are as follows:

$$\hat{\Gamma}^0 = \int d^3p \left[ b_1(p)\bar{b}_1(p) + b_2(p)\bar{b}_2(p) + b_3(p)\bar{b}_3(p) + b_4(p)\bar{b}_4(p), * \right],$$

$$\hat{\Gamma}^1 = \int d^3p \left[ b_1(p)b_4(p) - \bar{b}_4(p)\bar{b}_1(p) + b_2(p)b_3(p) - \bar{b}_3(p)\bar{b}_2(p), * \right],$$

$$\hat{\Gamma}^2 = i \int d^3p \left[ -b_1(p)b_4(p) - \bar{b}_4(p)\bar{b}_1(p) + b_2(p)b_3(p) + \bar{b}_3(p)\bar{b}_2(p), * \right],$$

$$\hat{\Gamma}^3 = \int d^3p \left[ b_1(p)b_3(p) - \bar{b}_3(p)\bar{b}_1(p) - b_2(p)b_4(p) + \bar{b}_4(p)\bar{b}_2(p), * \right], \tag{A4}$$

$$\hat{\Gamma}^4 = i\hat{\Gamma}^5 = i \int d^3p \left[ b_1(p)b_3(p) + \bar{b}_3(p)\bar{b}_1(p) + b_2(p)b_4(p) + \bar{b}_4(p)\bar{b}_2(p), * \right],$$

$$\hat{\Gamma}^6 = i \int d^3p \left[ b_1(p)b_2(p) + \bar{b}_2(p)\bar{b}_1(p) - b_3(p)b_4(p) - \bar{b}_4(p)\bar{b}_3(p), * \right],$$

$$\hat{\Gamma}^7 = \int d^3p \left[ b_1(p)b_2(p) - \bar{b}_2(p)\bar{b}_1(p) + b_3(p)b_4(p) - \bar{b}_4(p)\bar{b}_3(p), * \right].$$

Formulas of Lorentz invariant gamma operators of rotations $\hat{\Gamma}^{ab}$ are as follows:

$$\hat{\Gamma}^{01} = \int d^3p \left[ b_1(p)b_4(p) + \bar{b}_4(p)\bar{b}_1(p) + b_2(p)b_3(p) + \bar{b}_3(p)\bar{b}_2(p), * \right],$$

$$\hat{\Gamma}^{02} = -i \int d^3p \left[ b_1(p)b_4(p) - \bar{b}_4(p)\bar{b}_1(p) - b_2(p)b_3(p) + \bar{b}_3(p)\bar{b}_2(p), * \right],$$

$$\hat{\Gamma}^{03} = \int d^3p \left[ b_1(p)b_3(p) + \bar{b}_3(p)\bar{b}_1(p) - b_2(p)b_4(p) - \bar{b}_4(p)\bar{b}_2(p), * \right],$$

$$\hat{\Gamma}^{04} = i \int d^3p \left[ b_1(p)b_3(p) - \bar{b}_3(p)\bar{b}_1(p) + b_2(p)b_4(p) - \bar{b}_4(p)\bar{b}_2(p), * \right],$$

$$\hat{\Gamma}^{06} = i \int d^3p \left[ b_1(p)b_2(p) + \bar{b}_2(p)\bar{b}_1(p) - b_3(p)b_4(p) + \bar{b}_4(p)\bar{b}_3(p), * \right],$$

$$\hat{\Gamma}^{07} = \int d^3p \left[ b_1(p)b_2(p) + \theta(p)\bar{b}_1(p) + b_3(p)b_4(p) + \bar{b}_4(p)\bar{b}_3(p), * \right],$$

$$\hat{\Gamma}^{12} = -i \int d^3p \left[ b_1(p)\bar{b}_1(p) - b_2(p)\bar{b}_2(p) - b_3(p)\bar{b}_3(p) + b_4(p)\bar{b}_4(p), * \right],$$

$$\hat{\Gamma}^{13} = \int d^3p \left[ b_1(p)\bar{b}_2(p) - b_2(p)\bar{b}_1(p) + b_3(p)\bar{b}_4(p) - b_4(p)\bar{b}_3(p), * \right],$$

$$\hat{\Gamma}^{14} = i \int d^3p \left[ b_1(p)\bar{b}_2(p) + b_2(p)\bar{b}_1(p) + b_3(p)\bar{b}_4(p) + b_4(p)\bar{b}_3(p), * \right],$$

$$\hat{\Gamma}^{16} = i \int d^3p \left[ -b_3(p)\bar{b}_1(p) - b_1(p)\bar{b}_3(p) + b_4(p)\bar{b}_2(p) + b_2(p)\bar{b}_4(p), * \right],$$

$$\hat{\Gamma}^{17} = \int d^3p \left[ b_3(p)\bar{b}_1(p) - b_1(p)\bar{b}_3(p) - b_4(p)\bar{b}_2(p) + b_2(p)\bar{b}_4(p), * \right], \tag{A5}$$

$$\hat{\Gamma}^{23} = -i \int d^3p \left[ b_1(p)\bar{b}_2(p) + b_2(p)\bar{b}_1(p) - b_3(p)\bar{b}_4(p) - b_4(p)\bar{b}_3(p), * \right],$$

$$\hat{\Gamma}^{24} = \int d^3p \left[ b_1(p)\bar{b}_2(p) - b_2(p)\bar{b}_1(p) - b_3(p)\bar{b}_4(p) + b_4(p)\bar{b}_3(p), * \right],$$

$$\hat{\Gamma}^{26} = \int d^3p \left[ b_3(p)\bar{b}_1(p) - b_1(p)\bar{b}_3(p) + b_4(p)\bar{b}_2(p) - b_2(p)\bar{b}_4(p), * \right],$$

$$\hat{\Gamma}^{27} = i \int d^3p \left[ b_3(p)\bar{b}_1(p) + b_1(p)\bar{b}_3(p) + b_4(p)\bar{b}_2(p) + b_2(p)\bar{b}_4(p), * \right],$$

$$\hat{\Gamma}^{34} = i \int d^3p \left[ b_1(p)\bar{b}_1(p) - b_2(p)\bar{b}_2(p) + b_3(p)\bar{b}_3(p) - b_4(p)\bar{b}_4(p), * \right],$$

$$\hat{\Gamma}^{36} = i \int d^3p \left[ b_4(p)\bar{b}_1(p) + b_1(p)\bar{b}_4(p) + b_3(p)\bar{b}_2(p) + b_2(p)\bar{b}_3(p), * \right],$$

$$\hat{\Gamma}^{37} = \int d^3p \left[ -b_4(p)\bar{b}_1(p) + b_1(p)\bar{b}_4(p) - b_3(p)\bar{b}_2(p) + b_2(p)\bar{b}_3(p), * \right],$$

$$\hat{\Gamma}^{46} = \int d^3p \left[ b_4(p)\bar{b}_1(p) - b_1(p)\bar{b}_4(p) - b_3(p)\bar{b}_2(p) + b_2(p)\bar{b}_3(p), * \right],$$

$$\hat{\Gamma}^{47} = i \int d^3p \left[ -b_4(p)\bar{b}_1(p) - b_1(p)\bar{b}_4(p) + b_3(p)\bar{b}_2(p) + b_2(p)\bar{b}_3(p), * \right],$$

$$\hat{\Gamma}^{67} = -i \int d^3p \left[ b_1(p)\bar{b}_1(p) + b_2(p)\bar{b}_2(p) - b_3(p)\bar{b}_3(p) - b_4(p)\bar{b}_4(p), * \right].$$

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
