# Peer review of "Vacuum and Spacetime Signature in the Theory of Superalgebraic Spinors"

_universe, doi:10.3390/universe5070162_

Reviewer 1 Report

In this paper, a new formalism involving spinors and Dirac matrices is presented. Actually, new algebraic structures playing role of Dirac matrices are introduced on the base of Grssmann numbers. Further, various field theory constructions are defined with use of these structures.

While the idea is rather interesting, it clearly needs better presentation. First, some cumbersome formulas like Eq. (9) must be moved to Appendices. Second, actually only instruments for constructing the free field theory are given while it is interesting what Lagrangian can be constructed on the base of this formalism.  Third, a better motivation for the formalism proposed by authors is needed, especially, to answer what are advantages of this approach. Thus, I think that the paper needs more extension and clarification to answer these questions.

Author Response

Thanks to the reviewer for helpful comments.

English language has been corrected.

Structure of article has been corrected.

1. >First, some cumbersome formulas like Eq. (9) must be moved to Appendices.

- has been corrected.

2. >Second, actually only instruments for constructing the free field theory are given while it is interesting what Lagrangian can be constructed on the base of this formalism.

-   The main objective of the article was to study the structure of the vacuum and the space-time signature. It is of course interesting what Lagrangian can be constructed on the base of this formalism. This is a separate large study. However, a section was added to the article showing the possibilities of the developed approach for studying Clifford and spinor bundles.

3.>Third, a better motivation for the formalism proposed by authors is needed, especially, to answer what are advantages of this approach. I think that the paper needs more extension and clarification to answer these questions.

- Sections were added to the article, showing the necessity of transition from the theory of algebraic spinors to the proposed formalism. A section was also added showing the possibilities of the developed approach for studying Clifford and spinor bundles.

Reviewer 2 Report

The author develops a formalism that allows a representation of Dirac algebra in terms of grassmann variables and its derivatives. The presentation is adequate. The construction is based on articles and works of the same author. As far as I check the results are consistent, but it is not clear form me that this material can be of some utility for doing physics computations. It is just a formal material that can be of interest in mathematical physics. I recommend that the article could be published if the editors consdier that the material exposed is along the interest of the journal.

Author Response

Thanks to the reviewer for helpful comments.

English language has been corrected.

1.>The construction is based on articles and works of the same author.

- Structure of the article has been corrected. Sections were added to the article, showing the necessity of transition from the theory of algebraic spinors to the proposed formalism.

Added publications of authors whose works in the same field.

2.>it is not clear form me that this material can be of some utility for doing physics computations. It is just a formal material that can be of interest in mathematical physics.

- The main objective of the article was to study the structure of the vacuum and the spacetime signature. This is an important physical issue. The results obtained narrow down the possible directions of research, since they show that there are no more spaces with two or more timelike axes.

In addition, the proposed approach can also be used as a basis for further research in the field of noncommutative geometry and the study of the properties of spinors in a curved spacetime.

The developed mathematical formalism can be used in quantum field theory, including, possibly, to construct a quantum theory of gravity. The theory of Clifford algebras gives the most general description of the properties of spinors. Quantum field theory is based on the use of the method of second quantization. However, the approaches that were still available in the framework of the theory of algebraic spinors had nothing to do with the second quantization method. The proposed approach automatically leads to the appearance of second quantization.

To show some of the features of the proposed approach a section was added showing the possibilities of the developed approach for studying Clifford and spinor bundles.

Reviewer 3 Report

This paper takes a set of operators gamma^mu, constructed in a previous paper, which act on operators built from a set of four Grassman fields.  The author computes the action of these operators on something they call a "fermionic vacuum operator".  Additional operators Gamma^mu are written down (though the details of the construction are not made clear) which are claimed to be Lorentz invariant.  The author claims that the results preferentially pick out a (+1,-1,-1,-1) signature for spacetime.

There are many problems with this paper.  I will mention only a few.  First of all, the operators constructed should not be called gamma matrices; they do not obey a Clifford algebra.  In a Clifford algebra, one has {gamma^mu, gamma^nu} = eta^{mu nu}, and in particular any given gamma matrix squares to plus or minus one.  This ensures that the gamma matrices can only have eigenvalues that are fourth roots of unity.  The operators in the paper have zero eigenvalues, and are thus should not be identified with gamma operators.  In fact, looking at the actual construction of the operators, gamm^0 is the number operator for the Grassman fields, as the author notes later, while the other gamma operators are simply the six rotation matrices in the so(4) which acts on the Grassman fields, and so of course they don't obey a Clifford algebra.  Also, it is quite misleading to label them with a vector index mu, since there is no Lorentz group which acts sensibly on that index in a vector representation.

The construction of the creation and annihilation operators in (5) is strange.  The author seems to be saying that for any desired p, one can find a set of omega_{0k} which will give the right value, but doesn't say what those omega's should be (and it's also just not clear what the notation of (5) is meant to represent).

It is not explained in what sense the operator Psi_V should be thought of as a vacuum, nor what the significance is that some gamma's or combinations of gamma's annihilate this operator and some don't.  And finally, the claim that these considerations fix the space time signature (although they would seem to fix it to (+1,-1,-1,-1,-1,-1) rather than (+1,-1,-1,-1), even by the arbitrary rules being applied) seems to be far too strong a conclusion.

Author Response

Thanks to the reviewer for comments.

English language in the article has been corrected.

Structure of article has been corrected.

1.>the operators constructed should not be called gamma matrices; they do not obey a Clifford algebra.  In a Clifford algebra, one has {gamma^mu, gamma^nu} = eta^{mu nu}, and in particular any given gamma matrix squares to plus or minus one.  This ensures that the gamma matrices can only have eigenvalues that are fourth roots of unity.  The operators in the paper have zero eigenvalues, and are thus should not be identified with gamma operators. In fact, looking at the actual construction of the operators, gamm^0 is the number operator for the Grassman fields, as the author notes later, while the other gamma operators are simply the six rotation matrices in the so(4) which acts on the Grassman fields, and so of course they don't obey a Clifford algebra.

- the operators constructed should not be called gamma matrices, of course. That is why they are called gamma operators in the article, not gamma matrices. They are analogs of gamma matrices, but they turn out to be more complex objects. Gamma operators satisfy the Clifford algebra relations only when acting on an element of algebra, which is an analog of a column in the matrix theory.

In quantum field theory, the spinor field operator turns out to be a more complex object than the matrix column. The developed approach corresponds exactly to the description of the properties of field operators in the secondary quantization formalism.

In the revised version of the article much attention is paid to clarifying this issue. We introduced the concept of large Clifford algebra, which is the usual Clifford algebra, and on the basis of which all other constructions are constructed.

In addition, the concept of small Clifford algebra was introduced. It arises in the case when the number of basis vectors of a large Clifford algebra corresponds to the number of elements that a spinor-column should have. This is the algebra of operators acting on such a vector of large Clifford algebra. The generators of the small Clifford algebra are the gamma operators. They are commutators constructed from Grassmann variables and derivatives with respect to them. It is quite natural that such commutators can act on any elements of a large Clifford algebra, and not only on elements corresponding to components of spinors-columns.

2.>Also, it is quite misleading to label them with a vector index mu, since there is no Lorentz group which acts sensibly on that index in a vector representation.

- This remark is taken into account, and indices not related to the generalization of the Lorentz rotations are indicated in Latin letters in the revised version of the article. Unfortunately, such a replacement is not always possible, since the matrices Î“a are constructed on the basis of the matrices γa.

3.>The construction of the creation and annihilation operators in (5) is strange.  The author seems to be saying that for any desired p, one can find a set of omega_{0k} which will give the right value, but doesn't say what those omega's should be (and it's also just not clear what the notation of (5) is meant to represent).

- Formula (5) (in the revised version of article (26)) is given to clarify how annihilation and creation operators are related to Gramman densities and derivatives with respect to them.
We consider in (5) active Lorentz transformations. If there is one time-like axis, the parameters Ï‰0k are uniquely related to the direction of the resulting spatial impulse.

4.>It is not explained in what sense the operator Psi_V should be thought of as a vacuum

- Constructed operator  Î¨V satisfies all the requirements for the vacuum operator of quantum field theory. It is invariant under Lorentz transformations. And at the same time, it is the Clifford vacuum, since annihilation operators, when acted upon, give zero, and creation operators create single-particle states of the spinors.

5.>what the significance is that some gamma's or combinations of gamma's annihilate this operator and some don't. And finally, the claim that these considerations fix the space time signature (although they would seem to fix it to (+1,-1,-1,-1,-1,-1) rather than (+1,-1,-1,-1), even by the arbitrary rules being applied) seems to be far too strong a conclusion.

- The revised version of the article provides a more detailed explanation on this issue. It is shown that operators whose action on the vacuum is non-zero cannot have eigenvalues on single-particle state vectors. This means that these operators cannot correspond to the operators of physical measurable quantities. Among other things, this means that these operators cannot correspond to the operator γ0 of the Dirac theory of spinors, even if they are multiplied by i.

Of course, the conclusions made about the spacetime signature rely on some assumptions. First, it is the assumption that the proposed formalism is consistent with the principles of measurement of physical quantities in quantum mechanics. Secondly, this is the assumption that the spinor vacuum is isotropic. An indication of the need for these assumptions has been added to the revised version of the article.

Reviewer 4 Report

The manuscript is exploring the problem of the metric of the signature of spacetime. This problem has been investigated first by Stephen Hawking (early universe-> fluctuation of spacetime) and has stimulated interest since then. I do not have any issue with exploring the topic through a Clifford algebra approach as the author does. However, I have few remarks, which I hope, will help the author to make his manuscript suitable for the journal. First of all, I would rewrite the abstract. Make sure the abstract is accessible to all of us (physicists). I know the topic is interesting and it deserves to be explore, but not in the way it is presented. Second, I think the author should check the typos in the text. I give one example: Line 22 and Line 28 (""pseudo-maioran", which is very confusing). I would also simplify a bit the calculations. Lastly, please rewrite the conclusion to summarize your findings in a more suitable manner. Besides that, the author seems to know what he's talking about and I'll be happy to go over his manuscript one more time. 

Author Response

Thanks to the reviewer for helpful comments.

English language has been corrected.

Structure of article has been corrected.

1.>First of all, I would rewrite the abstract. Make sure the abstract is accessible to all of us (physicists). I know the topic is interesting and it deserves to be explore, but not in the way it is presented.

- the abstract has been rewritten.

2.>Second, I think the author should check the typos in the text. I give one example: Line 22 and Line 28 (""pseudo-maioran", which is very confusing).

- I apologize for the typos. In the revised version of the article I tried to correct typos.

3.>I would also simplify a bit the calculations.

- The most cumbersome formulas transferred to the Appendix.

In accordance with the wishes of other reviewers, sections with motivation for using the proposed approach were added, as well as a section showing the possibilities of using the developed mathematical formalism in the physics of quantum fields and spacetime.

4.>Lastly, please rewrite the conclusion to summarize your findings in a more suitable manner.

- I completely rewrote this section and tried in it to briefly summarize each section of the article. Perhaps this will be more successful than the previous one.

Round  2

Reviewer 1 Report

By my opinion, now the manuscript matches all requirements, all my comments were properly addressed, so, the manuscript can be published.

Author Response

>By my opinion, now the manuscript matches all requirements, all my comments were properly addressed, so, the manuscript can be published.

- Thank you for your feedback.

Reviewer 3 Report

The author has made extensive modifications to the article, and this has improved the presentation.  Unfortunately, my main concerns have not yet been addressed.  It's true that the constructed "small Clifford algebra" acts as the Clifford algebra Cl(1,6) when restricted to a very specific subspace (linear combinations of theta and theta-derivatives), but this should have anything to do with 3+1 dimensional Lorentz is unclear.  The relation between the omega_{0k} and p has not satisfactorily been clarified (for instance a Lorentz transformation never transforms p=0 into anything non-zero).  There does not seem to be any reason at all why gamma^4, gamma^6, and gamma^7 should be treated any differently than gamma^1, gamma^2, and gamma^3.  And finally, the same issue that indicates that the spatial gamma's can't act as gamma^0, since they don't "have eigenvalues on single-particle state vectors" would seem to also disqualify them from acting as spatial gammas, if Lorentz invariance is to be maintained.

Author Response

>The relation between the omega_{0k} and p has not satisfactorily been clarified (for instance a Lorentz transformation never transforms p=0 into anything non-zero). 

- clarification was done on page 6 of the corrected article. New text is marked by the green color.

Furthermore, in the section 11 is added proof that the corresponding part of the decomposition is a decomposition by momentum.

>There does not seem to be any reason at all why gamma^4, gamma^6, and gamma^7 should be treated any differently than gamma^1, gamma^2, and gamma^3.  

- this problem is discussed in the section 12.Discussion of the corrected article. The author does not claim to fully explain all the properties of the spacetime. However, the article shows that regardless of the interpretation of the gamma operators γ4, γ6 and γ7, only operator γ0 can have a signature of +1, and all others must have a signature of -1.

>And finally, the same issue that indicates that the spatial gamma's can't act as gamma^0, since they don't "have eigenvalues on single-particle state vectors" would seem to also disqualify them from acting as spatial gammas, if Lorentz invariance is to be maintained.

- As far as I understand this remark, you mean the impossibility of transforming the state vector using the Lorentz transformation, since the action of the rotation generator on the state vector of the vacuum does not give zero. I admit the presence of incorrect interpretation of some formulas in the original version of the article. In this regard, the article has been corrected - see section 7 in the corrected version.